# AutoStat: DSL-based Automated Statistical Modeling from Natural Language

## Abstract

Statistical modeling plays a critical role and is widely used in data analysis across diverse domains. Despite its importance, existing workflows remain cumbersome: they rely on fragmented programming environments and domain-specific probabilistic programming languages that are verbose and difficult to use, especially for non-experts. Although many efforts have been made toward automated statistical modeling, the methods still suffer from low accuracy, high computational cost, and heavy reliance on manual intervention. To address these challenges, we present *AutoStat*, a novel Domain-Specific Language (DSL)-based framework for automating statistical modeling. AutoStat leverages *StatModelDSL*, the first compact and structured DSL that specifies complete modeling tasks in a unified and portable form. AutoStat further enhances the automated process via interactive modeling by integrating two agents – StatModelChatbot, which interactively refines underspecified user requirements, and StatModelCopilot, which generates executable DSL programs. With StatModelChatbot clarifying intent and StatModelCopilot emitting executable DSL, AutoStat compiles and executes the specification end-to-end, delivering the complex statistical models directly from natural-language dialogue. We demonstrate that the proposed StatModelDSL affords both LLM amenability and practical usability: when instantiated with GPT-4o, it yields a **91.59%** reduction in error rate and a **5.89%** uplift in user preference over a Stan-based workflow. Meanwhile, AutoStat achieves a **100%** syntax correctness rate for DSL generation and a **98.76%** semantic passing rate, significantly surpassing previous methods. Our dataset, codes, and models will be publicly released upon acceptance.

## 1 Introduction

Statistical modeling (Box, 1976; Gelman et al., 1995) provides a principled framework for analyzing data by formalizing assumptions about the underlying data-generating process. It involves constructing probabilistic models that link observed data with underlying variables, enabling both interpretation and prediction. Statistical modeling is widely applied across domains such as economics (Sims, 2012; Shavell, 2004), biology (Kaplan & Meier, 1958; Armitage & Doll, 2004), and social sciences (Holland, 1986; Deegan Jr, 1979), where it helps researchers quantify uncertainty, test hypotheses, and make predictions. However, existing workflows for statistical modeling remain overly complex and unfriendly to users, as revealed by Gelman et al. (2020). On the one hand, probabilistic programming languages (PPLs) such as Stan (Carpenter et al., 2017) and PyMC (Patil et al., 2010) are syntactically verbose and often difficult to interpret. On the other hand, the overall workflow is fragmented: even fitting a simple hierarchical regression typically requires preprocessing data in a general-purpose language, writing dozens of lines of Stan code with explicit priors, and then switching back for post-processing.

To simplify the workflow and improve usability, many efforts (Li et al., 2024; Gouk & Gao, 2024) have been made to utilize the general knowledge from Large Language Model (LLM). Given a task description, these methods typically prompt an LLM to synthesize end-to-end code in a chosen PPL. Despite these attempts, we found that automated statistical modeling persistently suffers from low accuracy, high computational cost, and heavy reliance on intervention due to three issues: 1) Insufficient specifications. Statistical models require fine-grained specifications (*e.g.,* different variable distributions and constraints), yet verbose PPL syntax often leads LLMs to misinterpret

critical details; for example, as shown in Appendix A, the number of sampling steps is frequently misunderstood in the generated code, hindering the modeling process. 2) Fragmentation. Requiring LLMs to bridge heterogeneous environments (*e.g.,* data preprocessing in Python while inference in Stan (Carpenter et al., 2017)) often yields inconsistent outputs, resulting in multiple types of code that are lengthy and difficult to read. (Figure 5 provides an example). 3) Lack of portability. Models specified for one inference engine are not readily transferable across backends (*e.g.,* Stan (Carpenter et al., 2017) to PyMC (Patil et al., 2010)), resulting in limited cross-engine comparability as well as reduced validation capability. We identified that the root cause of the given limitations lies not only in the modeling capability of LLMs but, more importantly, in the modeling-language substrate itself: Natural language is very underspecified for statistical modeling, while existing PPLs are overly verbose, fragmented, and require substantial domain-specific expertise. Taken together, these issues highlight the need for an intermediate abstraction that is structured and concise, providing a middle ground between underspecified natural language and verbose PPLs.

Motivated by this, we present a novel automated statistical modeling framework, ***AutoStat***. At its core is *StatModelDSL*, a Domain-Specific Language (DSL) that represents complete statistical modeling tasks in a unified and portable modeling language form. This abstraction offers three main benefits: 1) Completeness and clarity: Every component of a task (data, parameters, model, inference, output) is explicitly represented, reducing ambiguity and improving human readability; 2) Unification and portability: A single DSL program can be compiled into multiple inference engines (*e.g.,* Stan, PyMC), eliminating the need to switch between environments and avoiding framework lock-in; and 3) LLM-friendliness: Its structured design makes fine-grained details explicit, enabling LLMs to generate more accurate and reliable programs (Tam et al., 2024) than with free-form PPL code. We validate these advantages in Section 4.5.

Since statistical modeling inherently requires fine-grained details that are difficult to specify in a single attempt, we design StatModelChatbot to guide users through dialogue, progressively grounding their intent into the structured components of StatModelDSL (*e.g.,* data, parameters, and model blocks), enabling users to complete modeling tasks with ease. AutoStat incorporates two agents to support the automation process (illustrated in Figure 1). *StatModelChatbot* interactively supplements incomplete or ambiguous user descriptions to finalize all necessary task components, while *StatModelCopilot* translates the clarified descriptions into executable StatModelDSL programs. Together, these components allow users to specify and execute complex statistical models directly from natural language, while ensuring that the resulting code remains precise, portable, and reliable.

We demonstrate that StatModelDSL affords both LLM amenability and practical usability: when instantiated with an LLM, *e.g.,* GPT-4o, it yields a 91.59% reduction in error rate and a 5.89% uplift in user preference over a Stan-based workflow. Building on this DSL, AutoStat attains a 100% syntax-correctness rate for DSL generation and a 98.76% semantic passing rate, significantly surpassing prior methods. Our dataset, code, and models will be released upon acceptance.

In summary, our contributions include the following.

- We present AutoStat, an end-to-end DSL-based framework that converts natural language into executable, reproducible statistical workflows. At its core is StatModelDSL—a concise, unified, and standardized Domain-Specific Language that formalizes task specifications for automated statistical modeling.

- We enable interactive statistical modeling that iteratively aligns users' intent with StatModelDSL through two agents that converse to capture and refine intent and then effectively produce the specification into executable DSL programs—improving accuracy, usability, and efficiency.

- Extensive experiments on diverse statistical tasks show that the StatModelDSL-equipped AutoStat delivers consistently higher accuracy, reliability, and reproducibility, outperforming the SOTA automated modeling approaches.

## 2 RELATED WORK

### 2.1 STATISTICAL MODELING WORKFLOW

Current workflows for statistical modeling (Gelman et al., 2020) heavily rely on Probabilistic Programming Languages (PPLs)(van de Meent et al., 2018; Krapu & Borsuk, 2019), such as

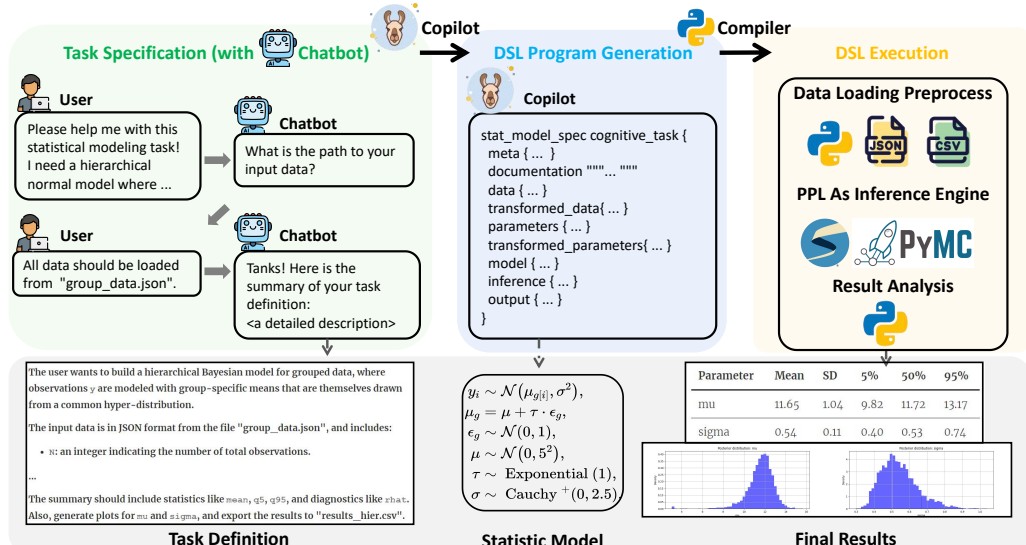

Figure 1: The workflow of AutoStat: StatModelChatbot first assists users in refining task details, after which StatModelCopilot generates the corresponding StatModelDSL program. The DSL compiler then executes this program to produce the final results.

Stan(Carpenter et al., 2017), PyMC (Patil et al., 2010), Turing.jl (Ge et al., 2018), and Pyro (Bingham et al., 2018). As noted by Gelman et al. (2020), while PPLs are powerful for statistical modeling, the complete workflow still requires external tools (e.g., Python or R) for data processing and analysis. Early attempts at automation, such as Fischer & Schumann (2003), transformed statistical models into automated data analysis pipelines, but the input rules were overly restrictive and the modeling expressiveness too limited for modern PPL environments. More recently, approaches leveraging LLMs (Gouk & Gao, 2024; Li et al., 2024) have emerged to translate natural language descriptions into PPL programs. However, these methods still suffer from fragmented environments and limited accuracy. In this work, we introduce a DSL-based framework that unifies the entire workflow. Leveraging LLMs, we make the workflow easy and reliable.

## 2.2 DOMAIN-SPECIFIC LANGUAGE FOR LARGE LANGUAGE MODELS

A Domain-Specific Language (DSL) (Fowler, 2010; Mernik et al., 2005) is a programming language tailored to a particular domain (*e.g.,* Markdown, SQL). Its simplified and structured design makes task specification clearer for humans and more reliable for LLMs. DSLs can be broadly divided into two categories: 1) Standalone DSLs, which define their own syntax and compiler infrastructure. Examples include MoVer (Ma & Agrawala, 2025), a motion verification language for controllable motion generation, and SPCC (Li et al., 2025c), a specialized language for CAD modeling. 2) Embedded DSLs, which are implemented within a host language such as Python. Examples include Liang et al. (2022), who encode robot policies as Python programs for LLM-based policy control, Makatura et al. (2025), who design a DSL to capture metamaterials in a structured and expressive form, and Li et al. (2025a), who develop a Python-based DSL for material modeling and employ vision-language models for code generation. In this work, we propose StatModelDSL, a standalone DSL that models the entire statistical modeling workflow, and we use LLMs to automatically generate programs in this DSL.

## 3 METHOD

To address the verbosity of existing PPLs and enable a user-friendly, end-to-end automated statistical modeling workflow, we propose *AutoStat*. As illustrated in Figure 1, AutoStat leverages the concise and structured StatModelDSL (Section 3.1) to represent the entire statistical modeling task, replacing traditional PPLs. The StatModelChatbot (Section 3.2) interactively collects and clarifies task details from the user, after which the StatModelCopilot (Section 3.4) accurately generates specific DSL programs and executes them. With AutoStat, even novice users can complete complex statistical modeling tasks using only natural language.

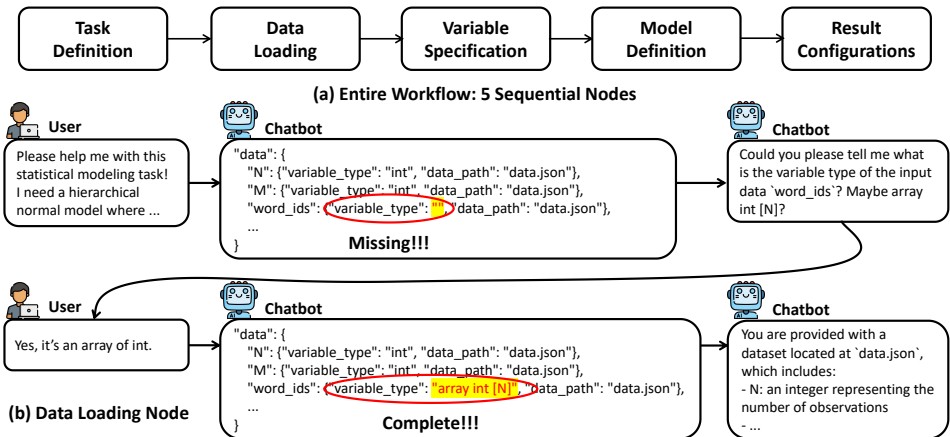

Figure 2: An illustration of the StatModelChatbot workflow. The chatbot leverages the user's description to populate a predefined schema. If the provided information is insufficient to complete all required fields, the chatbot prompts the user for additional input until the schema is fully specified.

## 3.1 STATMODELDSL

We present StatModelDSL, a Domain-Specific Language that systematically expresses statistical modeling tasks in a structured and interpretable way. It integrates all components of the workflow, from data pre-processing and model specification to inference and result analysis, and can be compiled directly into executable programs across different probabilistic programming environments (*e.g.,* Stan, PyMC). The design of StatModelDSL emphasizes three key characteristics:

- **Clarity**: Each component of a statistical model is explicitly organized into separate blocks, ensuring that fine-grained details are represented in a transparent and structured manner.
- **Completeness**: The DSL captures the entire modeling pipeline, while our compiler automates execution end-to-end, reducing the fragmented and redundant steps of traditional workflows.
- **Portability**: The DSL can be translated into multiple inference engines, avoiding framework lock-in and enhancing flexibility. This cross-platform capability also facilitates the design of LLM-based agents, as they can generate a single unified DSL program without needing to handle environment-specific differences.

When executing the DSL, we first parse the code into an Abstract Syntax Tree (AST) using the Lark parser (Shinan, 2021), which captures the hierarchical structure of the program and facilitates traversal and manipulation. We then leverage a general-purpose programming language (Python) to handle data loading, preprocessing, and post-processing tasks such as plotting and result exporting, while the core statistical model is executed using the target PPL environment (*e.g.,* Stan or PyMC) to complete the end-to-end workflow. Appendix B provides a comprehensive description of StatModelDSL's design, execution process, and illustrative examples, as well as demonstrations of its portability by compiling the program into different PPL backends.

## 3.2 STATMODELCHATBOT

Statistical modeling is inherently detail-intensive, requiring specifications such as variable types and distributional parameters. Since it is difficult for users to provide a complete description in a single attempt, we design a supportive chat agent (Wolf et al., 2019; Adiwardana et al., 2020), StatModelChatbot, that interactively assists users in refining task details, thereby ensuring completeness.

To ensure that the chatbot systematically verifies the presence of all necessary task information and produces outputs in a stable format, inspired by (Caufield et al., 2024; Lu et al., 2025; Shiri et al., 2024), we design a schema that specifies the key elements the chatbot must extract. Furthermore, to improve accuracy, following (Zhou et al., 2022), we decompose the task into five sequential nodes: task definition, data loading, variable specification, model definition, and result configuration.

As illustrated in Figure 2. For a certain node, the chatbot extracts key information from the user's prompt to populate a pre-designed schema that specifies both required elements and optional ones. If

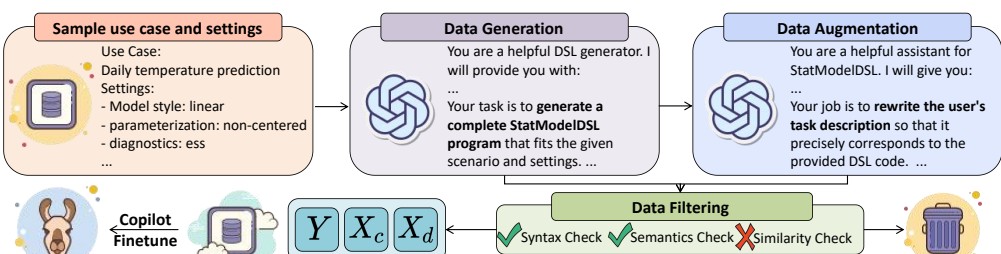

Figure 3: The pipeline of the StatModelDataset construction process, which contains three main steps: data generation, data augmentation, and data filtering.

all required fields are successfully filled, the node is considered complete, and the chatbot generates a natural-language summary for user confirmation before proceeding to the next node. If critical information is missing, the chatbot gives feedback to the user to provide the necessary details, and the process repeats. At the end of this interaction, we obtain a complete schema containing all essential specifications, along with a finalized natural-language task definition.

## 3.3 STATMODELDATASET

To effectively train and evaluate LLMs on our DSL, it is crucial to develop a diverse, and high-quality dataset. To this end, we construct StatModelDataset, whose construction pipeline, illustrated in Figure 3, comprises three stages: data generation, data augmentation, and data filtering.

### 3.3.1 DATA GENERATION

Due to the lack of existing datasets for statistical modeling, we leverage the strong in-context learning capabilities of LLMs (Brown et al., 2020; Yang et al., 2024; Li et al., 2025b) to generate synthetic data. To ensure the diversity of our dataset, we predefine 56 domains (*e.g.,* Time Series and Forecasting, Economics and Finance), each containing more than 10 potential use cases, resulting in a total of 590. In addition, we predefine a variety of model settings. For each use case, we randomly sample model settings and prompt GPT (Hurst et al., 2024) to generate both a concise task description $X_c$ and the corresponding DSL code $Y$. To guarantee generation quality, we also provide LLMs with a natural-language description of the DSL syntax along with several detailed examples. Through this design, we are able to efficiently generate diverse base data.

### 3.3.2 DATA AUGMENTATION

A solely concise description is insufficient to capture the complexity of a statistical task; thus, we refine the initial task descriptions to provide greater detail. Specifically, we supply the DSL code to the LLM, along with illustrative examples, prompting it to generate a more comprehensive task description $X_d$. In this way, we construct a triplet dataset consisting of a concise task description $X_c$, a detailed task description $X_d$, and the corresponding DSL code $Y$. More details are illustrated in Appendix E.

### 3.3.3 DATA FILTERING

To ensure the quality of our dataset, the most critical step is filtering out erroneous and low-quality data. To this end, we designed the following data cleaning procedures tailored to our DSL.

- **Syntax check.** To ensure the syntax correctness of our DSL programs, we employ our DSL compiler to verify whether each DSL instance conforms to the syntax specification, while also checking for issues such as variable name reuse and other violations of syntactic rules.
- **Semantics check.** To assess whether each description is sufficiently complete and fully aligned with the DSL, we further prompt an LLM to verify the semantic consistency between them. This process ensures that the description does not omit critical details and that the DSL and the detailed description are matched at a fine-grained level. The prompt is shown in Appendix G.2
- **Similarity check.** To ensure dataset diversity and prevent the repetition of highly similar samples, we apply TF-IDF vectorization Salton et al. (1975) to all DSL code and remove instances that exceed a similarity threshold, thereby filtering out overly redundant code at the string level.

Although such strict filtering further reduces the dataset size, it ensures high-quality and reliable data, thereby providing a solid foundation for the subsequent training of our StatModelCopilot.

For evaluation, the final test set (Table 7) contains 323 high-quality items, categorized into three levels of complexity (simple, medium, and complex) based on the description $X_d$ length. Both training and testing datasets are verified by humans to ensure high quality.

### 3.4 STATMODELCOPILOT

After all task details are specified, an agent is responsible for converting the natural language description into a fully executable DSL program, enabling end-to-end automation. LLMs have demonstrated strong capabilities in code generation (Chen et al., 2021; Jiang et al., 2024; Hui et al., 2024; Guo et al., 2024), but for a new DSL that is both detail-intensive and instruction-heavy, relying solely on in-context learning (Dong et al., 2024) is insufficient. To address this challenge, we train our StatModelCopilot, which can follow complex statistical modeling instructions and generate syntactically correct DSL programs that strictly adhere to the specifications.

To maximize the effectiveness of our dataset and ensure that the model both learns the syntax of our DSL and faithfully follows complex user specifications, we adopt a curriculum learning approach (Bengio et al., 2009) with two training stages. In the first stage, we use concise task descriptions as input prompts, with the corresponding DSL code as labels. This stage focuses on teaching the LLM the syntax and structural rules of StatModelDSL, ensuring syntactic correctness. In the second stage, we extend training to detailed task descriptions, where the model must capture and follow all specified details, producing DSL programs that are not only syntactically valid but also strictly aligned with the input requirements. We leverage the instruction tuning method to train our models. The training losses are:

$$\mathcal{L}_1 = -\sum_{i=1}^{N_1} \log P(Y_i|(X_c)_i), \quad \mathcal{L}_2 = -\sum_{i=1}^{N_2} \log P(Y_i|(X_d)_i), \tag{1}$$

where $N_1, N_2$ denote the number of training samples for two stages, respectively.

## 4 EXPERIMENTS

We evaluate the proposed AutoStat framework to answer the following research questions:

- **RQ1**: How does AutoStat compared to other LLM-based methods?

- **RQ2**: What is the contribution to the designs in AutoStat (*i.e.,* two-stage training strategy, chatbot assistance) to the performance?

- **RQ3**: How does the base LLM affect the performance of our AutoStat?

- **RQ4**: As the foundation of AutoStat, does StatModelDSL provide advantages over traditional PPLs in LLM-based code generation and user usability?

- **RQ5**: Whether AutoStat can achieve strong performance and play a practical role in real-world statistical modeling tasks.

All experiments are conducted on our test set mentioned in 3.3. During testing, the detailed task description $X_d$ is used as the input prompt, and the target StatModelDSL program $Y$ serves as the ground truth statistical modeling program.

### 4.1 EXPERIMENTAL SETTINGS

**Baselines.** Following prior work, we evaluate AutoStat using GPT-4o and GPT-4o-mini (Hurst et al., 2024; Li et al., 2024), as well as Llama3-8B (Dubey et al., 2024; Gouk & Gao, 2024), under a few-shot learning setup (Brown et al., 2020; Parnami & Lee, 2022), where models are given the DSL specification and a few examples along with the task description. In contrast, our StatModelCopilot directly generates the DSL program from the task description alone.

Table 1: Performance comparison between our AutoStat and GPT-based baselines. "*" indicates prompting with two in-context examples. The best performance is highlighted **in bold**.

| Model | Syntax ↑ | | | | Semantics ↑ | | | |
|---|---|---|---|---|---|---|---|---|
| | Simple | Medium | Complex | Avg. | Simple | Medium | Complex | Avg. |
| Llama3-8B | 79.01 | 61.82 | 33.77 | 59.44 | 8.64 | 32.73 | 11.69 | 21.67 |
| Llama3-8B* | 90.12 | 76.97 | 58.44 | 75.85 | 59.26 | 51.52 | 40.26 | 50.77 |
| GPT-4o-mini | 98.77 | 96.36 | 88.31 | 95.05 | 90.12 | 82.42 | 48.05 | 76.16 |
| GPT-4o-mini* | 98.77 | 97.58 | 93.51 | 96.90 | 93.83 | 93.94 | 79.22 | 90.40 |
| GPT-4o | 97.53 | 99.39 | 96.10 | 98.14 | 92.59 | 93.33 | 77.92 | 89.47 |
| GPT-4o* | 98.77 | 100.00 | 96.10 | 98.76 | 96.30 | 95.15 | 81.82 | 92.26 |
| AutoStat (1B) | 98.77 | 97.58 | 92.21 | 96.59 | 91.36 | 90.30 | 68.83 | 85.45 |
| AutoStat (3B) | 98.77 | 96.36 | 98.70 | 97.52 | 93.83 | 92.73 | 77.92 | 89.47 |
| AutoStat (8B) | 100.00 | 100.00 | 100.00 | **100.00** | 98.77 | 99.39 | 97.40 | **98.76** |

**Metrics.** To evaluate the accuracy of our automation pipeline, following (Chen et al., 2021; Kulal et al., 2019; Guo et al., 2025), we employ Pass@1 to assess the DSL code generation success rate. Specifically, we compare the generated DSL with the ground-truth DSL at both the AST level and after conversion to the target PPL code. A sample is considered successful only if all components, including model design, data processing, and variable specifications, exactly match the ground truth.

**Implementation Details.** We fine-tune Llama3-(1B, 3B, 8B) (Dubey et al., 2024) using the LoRA framework (Shen et al.) on a single NVIDIA A40 GPU. More details are shown in Table 6.

## 4.2 RESULTS ANALYSIS (RQ1)

Table 1 presents a comprehensive comparison between StatModelCopilot and all baseline methods. We summarize our key observations as follows:

- Our fine-tuned 8B Copilot achieves the best performance across tasks of varying difficulty, excelling at both the syntax and semantic levels. Specifically, it attains a **100% passing rate on syntax checks** and a **98.76% passing rate on semantic checks**. These results indicate that StatModelCopilot can accurately understand user requirements, capture nearly all critical task details, and generate programs that comply with our DSL specification. Compared to the weak performance of the pre-trained Llama baselines, the superior performance further validates the effectiveness of our training data and methodology.

- Providing additional examples significantly improves model performance. With just two examples, GPT-4o, GPT-4o-mini, and Llama3-8B all show substantial gains at the syntax and semantic levels. This demonstrates the strong in-context learning ability of powerful LLMs: supplementary examples help them better understand and apply our DSL, leading to more accurate task execution.

- Model capacity plays a crucial role in the performance of statistical modeling tasks, particularly in semantic understanding and handling complex tasks. In our complex-level semantic evaluations, GPT-4o outperforms GPT-4o-mini by nearly 30%. Similarly, our 8B StatModelCopilot achieves around 30% higher accuracy than its 1B counterpart and about 20% higher than the 3B version. These results suggest that for tasks requiring fine-grained semantic comprehension and precise specification, final performance strongly depends on the underlying model capacity, regardless of whether fine-tuning or prompt engineering is applied.

## 4.3 ABLATION STUDIES (RQ2)

To assess the effectiveness of our two-stage learning strategy and the StatModelChatbot, we perform ablation studies on the 8B StatModelCopilot: (1) "w/o Stage-One": training only with detailed descriptions $(X_d, Y)$; (2) "w/o Stage-Two": training only with concise descriptions $(X_c, Y)$; (3) "w/o Chatbot": replacing the standardized prompts generated by StatModelChatbot with masked, user-toned prompts $X_d$ to simulate users' prompts to examine its contribution to the overall workflow.

From the results shown in Table 2, we observe that:

Table 2: Effect of our training strategy and the StatModelChatbot.

| Model | Syntax ↑ | | | | Semantics ↑ | | | |
|---|---|---|---|---|---|---|---|---|
| | Simple | Medium | Complex | Avg. | Simple | Medium | Complex | Avg. |
| Ours | 100.00 | 100.00 | 100.00 | **100.00** | 98.77 | 99.39 | 97.40 | **98.76** |
| - w/o Stage-One | 98.77 | 100.00 | 96.10 | 98.76 | 96.30 | 95.15 | 81.82 | 92.26 |
| - w/o Stage-Two | 100.00 | 100.00 | 98.79 | 96.28 | 81.48 | 61.21 | 41.56 | 61.61 |
| - w/o Chatbot | 97.53 | 97.58 | 94.81 | 96.90 | 59.26 | 50.91 | 35.06 | 49.23 |

Table 3: Comparison across different base LLMs. For syntax and semantics, we report pass@1. For speed, we report average seconds per batch (s/batch), and for memory, we report GPU VRAM usage. The best results are highlighted **in bold**.

| Model | Size | Syntax ↑ | Semantics ↑ | Speed ↓ | Memory ↓ |
|---|---|---|---|---|---|
| Llama3 | 8B | **100.00** | **98.76** | 19.72 | **41.22** |
| Qwen3 | 8B | 99.38 | 96.59 | 23.64 | 41.32 |
| Qwen2.5-Coder | 7B | 98.76 | 94.43 | **19.22** | 41.64 |
| Mistral | 7B | 99.69 | 92.26 | 27.44 | 41.52 |

• In the first stage, after training the LLM with concise task descriptions, the results show that the model already achieves a strong syntax passing rate. This indicates that Stage-One successfully enables the LLM to grasp the grammatical rules of our DSL, as expected. However, its relatively weaker performance at the semantic level suggests that such training alone is insufficient for the model to capture key information from task descriptions or to fully understand the user's intent.

• Removing Stage-One training and directly learning from complex tasks affects performance, as the model struggles to acquire the syntax and semantics of StatModelDSL from limited data. A curriculum strategy—starting with simpler tasks and gradually increasing complexity—enables the model to internalize DSL structures and capture semantic intent more effectively, consistent with prior findings on curriculum learning (Bengio et al., 2009; Elman, 1993; Xu et al., 2020).

• Without the standard and complete prompt generated by StatModelChatbot, the performance of our StatModelCopilot drops dramatically. On the one hand, user descriptions may be incomplete or underspecified, making it difficult for the Copilot to generate a DSL program that exactly matches the target. On the other hand, while the training inputs $X_d$ follow standardized expressions, real user inputs are often more varied and irregular in format, which weakens the Copilot's performance. This mismatch is also reflected in the observed drop in syntax-level performance.

## 4.4 EFFECT OF BASE LLMS (RQ3)

To assess the impact of the underlying base model in StatModelCopilot, we conduct a comprehensive comparison across LLMs of similar size. In addition to Llama3-8B (Dubey et al., 2024), we evaluate Qwen3-8B (Yang et al., 2025), Qwen2.5-Coder-7B (Hui et al., 2024), and Mistral-7B (Jiang et al., 2023), providing a systematic analysis of how different architectures affect performance.

We evaluate both the syntax and semantics Pass@1 across the entire dataset. In addition, we assess inference cost and efficiency. Specifically, we adopt vLLM (Kwon et al., 2023) with a fixed batch size of 32, and measure the average inference time per batch and the GPU memory consumption.

As shown in Table 3, we have the following observations:

• Llama3 achieves the best overall performance both in terms of passing rate and memory usage.

• For models with similar sizes, the memory usage is roughly comparable; however, we observe clear differences in inference speed. These differences primarily stem from architectural design choices as well as the average generation length.

• We also test Qwen2.5-Coder to see if a code-specialized model would perform better, but it shows no clear advantage. This may be because it lacks additional training on probabilistic programming languages (Hui et al., 2024), which are central to our tasks.

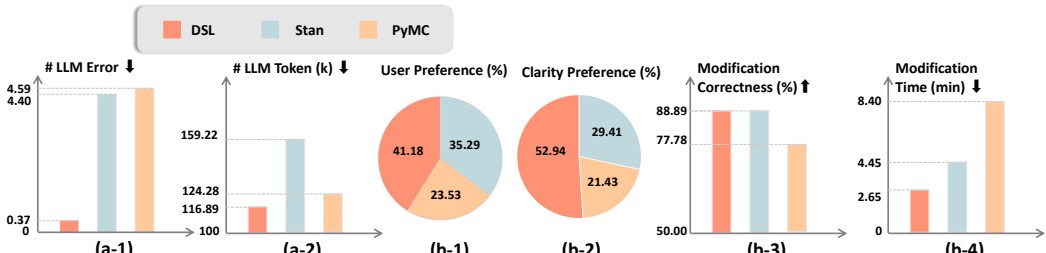

Figure 4: Performance comparison between the StatModelDSL-based workflow and PPL-based workflows. Subfigures (a-1) and (a-2) report experiments evaluating LLM-friendliness, while (b-1) to (b-4) present experiments evaluating user-friendliness.

## 4.5 STATMODELDSL EVALUATION (RQ4)

### 4.5.1 EXPERIMENTAL SETTINGS

We conduct both a quantitative evaluation and a user study to demonstrate that our StatModelDSL-based workflow is more accurate, efficient, and user-friendly compared to Stan and PyMC.

**Quantitative evaluation.** We uniformly prompt GPT-4o (Hurst et al., 2024) to generate programs on the test set under different environments. For fairness, when using StatModelDSL, we additionally provide its concise syntax specification to the model. We then evaluate the generated outputs along two dimensions: accuracy and cost. For accuracy, since implementations across different environments cannot be directly compared with rule-based checks, we adopt an LLM-as-a-judge approach (Liu et al., 2023; Zheng et al., 2023) (see Appendix G.2), where GPT-4o is asked to identify and count inconsistencies between the generated program and the task description. For cost, we measure the total number of tokens generated during inference on the entire dataset.

**User study.** To assess usability, we invite participants to modify programs under different environments to be familiar with different programming languages. Afterward, they are asked which environment they find most user-friendly, which they would prefer to use in the future, and which code representation they consider clearest and most readable. More details are shown in Appendix G.3.

### 4.5.2 RESULTS ANALYSIS

As shown in Figure 4, we have the following observations:

- As shown in (a-1) and (a-2), leveraging StatModelDSL dramatically improves the accuracy of GPT-4o: the error rate is reduced by more than **92.05%** compared to PPL-based workflows. In addition, token consumption is significantly lower, indicating that StatModelDSL is not only more accurate but also more efficient, making it a better fit for LLM-driven statistical modeling tasks.
- As shown in (b-1) and (b-2), most participants found our StatModelDSL programs clearer and more readable, and expressed a preference for using our DSL in similar tasks, demonstrating that StatModelDSL is genuinely user-friendly, benefiting from its clear and structured design.
- More specifically, results from (b-3) and (b-4) show that novice users achieved relatively high success rates when modifying both DSL and Stan programs. In addition, in terms of the time required for modification, DSL outperformed both Stan and PyMC by a clear margin, demonstrating that StatModelDSL is particularly user-friendly for novices, making code modification simpler and more efficient while also enhancing overall readability.

## 4.6 REAL-WORLD EVALUATION (RQ5)

To further evaluate the effectiveness of our AutoStat pipeline in real-world scenarios, we constructed two datasets of statistical modeling tasks:

- **Textbook-derived tasks.** We randomly curated 50 Bayesian modeling tasks from the official Stan Examples Repository[1], drawing exclusively from three authoritative textbooks in Bayesian statistics Gelman & Hill (2007); Lee & Wagenmakers (2014); Kéry & Schaub (2011).
- **Research paper–derived tasks.** To assess the utility of StatModelDSL in contemporary research settings, we analyzed 83 papers published in Bayesian Analysis[2] (2020–present). After automated

---

[1] https://github.com/stan-dev/example-models
[2] https://projecteuclid.org/journals/bayesian-analysis

extraction and subsequent manual filtering, we selected 50 statistical modeling tasks with clear and well-defined experimental descriptions.

All experimental settings follow exactly the same configuration as described in Section 4.5. We compare the AutoStat pipeline against the traditional workflow based on Python and Stan, evaluating their respective error rates and recording the syntax correctness rate of our DSL.

From the results shown in Table 4, we can observe that: 1) **Our AutoStat system is capable of handling a wide range of real-world scenarios.** It can successfully complete relatively straightforward data analysis tasks derived from textbooks, as well as more complex statistical modeling tasks that appear in cutting-edge research. 2) The formal task specification provided by StatModelDSL **makes statistical modeling tasks clearer and more explicit**, re-

Table 4: Comparison of error rates between AutoStat and the baseline. "Syntax" refers to the passing rate of syntax checks performed by our DSL parser, and "Error" denotes the number of inconsistencies between the generated output and the task requirements as judged by the LLM.

|  | Textbook | | Research paper | |
| --- | --- | --- | --- | --- |
|  | Syntax (%) | # Error | Syntax (%) | # Error |
| Python + Stan | - | 0.00 | - | 8.90 |
| AutoStat | 100 | 0.00 | 100 | 5.48 |

sulting in a lower error rate. Every detail of the task is fully represented, enabling our system to perform more effectively, especially on complex modeling problems. Moreover, our StatModelChatbot enables interactive communication with users, allowing the system to fully understand task requirements. Even novice users with limited background in statistical modeling can easily use the system to construct and execute sophisticated statistical experiments.

For these evaluations, we will release all testing data, task descriptions, DSL implementations, and detailed metadata—including the source papers and the exact locations of the corresponding experiments within those papers—together with our full system. Additional examples and further details can be found in the Appendix H.

## 5 CONCLUSION AND FUTURE WORK

In this work, we propose AutoStat, a unified framework for automating statistical modeling. AutoStat leverages StatModelDSL, the first Domain-Specific Language designed to simplify the entire workflow of statistical modeling. Building on this foundation, AutoStat also consists of StatModelChatbot and StatModelCopilot, which enable end-to-end automation of the task. With our framework, even novice users can complete complex statistical modeling tasks simply by delivering the complex statistical models directly from natural-language dialogue. Through extensive experiments, we demonstrate that our method improves both accuracy and usability.

Looking forward, our framework offers potential for further scaling and improvement. StatModelDSL can be extended to support richer data structures, more flexible data input functions, and advanced output processing. The StatModelCopilot can be further enhanced by incorporating real-world data for both training and evaluation, ensuring more reliable performance. Additionally, developing a more lightweight version of the model (*e.g.,* using MoE-based architectures) would improve efficiency and deployability for users. We also envision an online platform that allows users to access the full workflow without requiring local GPU resources, making automated statistical modeling more widely accessible.

## REPRODUCIBILITY STATEMENT

Our code, dataset, and model weights will be publicly released upon acceptance. All implementation details and training cost are shown in Appendix F Table 6. Our dataset, codes, and models will be publicly released upon acceptance.

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

## USE OF LLMS IN OUR WORK

In this paper, we leverage LLMs in several ways: (1) polishing the writing of our manuscript, (2) retrieving relevant papers, (3) refining the design of prompts, and (4) generating part of our dataset (as described in Section 3.3). Importantly, all outputs produced by LLMs were carefully reviewed and verified by humans to ensure accuracy and reliability.

## CONTENTS

## A   ENVIRONMENTS COMPARISON: AN EXAMPLE

Figure 5: We prompt GPT-4o to generate code from the same task description under different probabilistic programming environments.

As illustrated in Figure 5, we instructed GPT-4o to generate code under different probabilistic programming environments using the identical detailed task description. Through this experiment, three key observations can be explicitly drawn:

- The Stan-generated code is notably lengthy, and there exists a stark discrepancy in coding style between Stan and Python, leading to weak readability. Additionally, the overall implementation is highly complex and cumbersome.

- While the PyMC code is relatively concise, it suffers from poor clarity and poses a steep learning curve for beginners. For instance, in this task, we require the "sigma_latent" variable to follow a Cauchy distribution with the constraint of being greater than 0. In PyMC, this necessitates the specific use of "HalfCauchy"—an operation that beginners may struggle to implement straight-forwardly.

- In contrast, our proposed DSL achieves both conciseness and clarity. It streamlines numerous repetitive yet essential procedures (*e.g.,* plotting and data loading) into single, unambiguous lines of code. Furthermore, the statistical model component of our DSL is remarkably intuitive: for the "sigma_latent" variable mentioned above, we simply apply the constraint "$> 0$" for modification. This approach also affords greater flexibility to the model formulation.

We further instructed GPT-4o to act as a critic, tasked with identifying all discrepancies between the generated code and the task description. The key findings are as follows:

- Regarding the Stan code, there is an error in the sampling iterations. The task description explicitly requires sampling 2000 iterations with 1000 warmup iterations, yet the generated code fails to adhere to this specification.

- For the PyMC code, a mistake is present in the results saving process. As specified in the task, the target metric to be saved is q95", but the generated code incorrectly saves hdi_97.5" instead.

- In contrast, the code generated using our DSL is entirely accurate. GPT-4o did not detect any mismatches between the DSL code and the task definition.

# B   STATMODELDSL

## B.1   DSL COMPONENTS

Our proposed StatModelDSL consists of several building blocks, each serving a specific role in statistical modeling. The overall structure is as follows:

```
stat_model_spec <model_name> {
    meta { ... }
    documentation "..." | """..."""
    data { ... }
    transformed_data { ... } (Optional)
    parameters { ... }
    transformed_parameters { ... } (Optional)
    model { ... }
    inference { ... }
    output { ... }
}
```

Block Descriptions:

- **meta**: Specifies metadata such as inference engine, author, and version information.

- **documentation**: Provides human-readable notes or descriptions of the program in natural language.

- **data**: Defines input data sources and corresponding constraints.

- **transformed_data**: (Optional) Declares new variables derived from input data.

- **parameters**: Lists model parameters with constraints and prior distributions.

- **transformed_parameters**: (Optional) Defines transformed parameters as variables for down-stream modeling.

- **model**: Contains the core statistical model specification.

- **inference**: Configures inference algorithms and settings.

- **output**: Specifies outputs, including monitored parameters, summaries, diagnostics, plots, and export options.

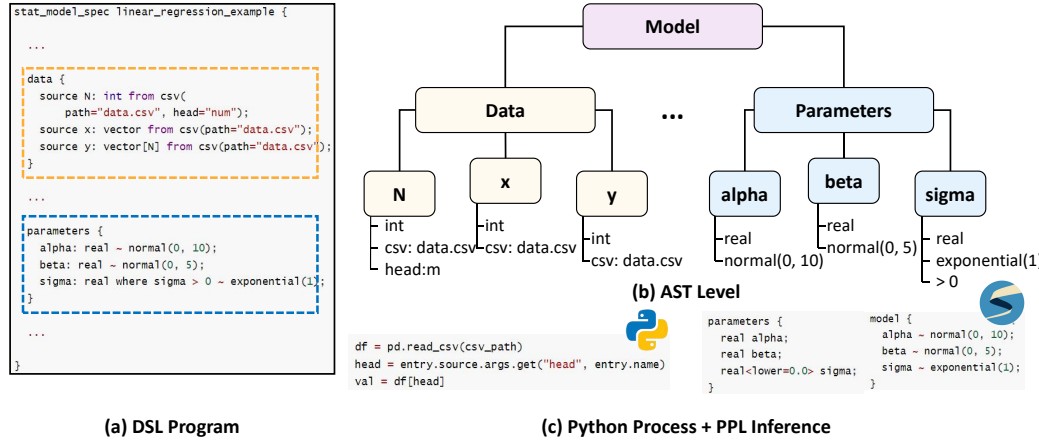

Figure 6: Our StatModelDSL execution pipeline.

## B.2 DSL EXECUTION

As illustrated in Figure 6, our StatModelCompiler takes two steps to execute a DSL program.

- **(a) DSL → (b) AST.** For a DSL program, we first parse the code using the Lark parser, converting it into a hierarchical tree structure with the format "program → blocks → entries". This tree representation facilitates program comprehension and allows us to read, validate, and process the code in a structured, block-wise manner.
- **(b) AST → (c) Python+PPL.** At the AST level, we handle each block differently. For example, the data block is used to load and validate input data according to its specifications, while the parameters block is converted into the corresponding code in the target PPL.

This is the execution EBNF of StatModelDSL, which formally specifies the rules for parsing and compiling the entire modeling workflow.

```
Execution EBNF of StatModelDSL

start: stat_model_spec

stat_model_spec: "stat_model_spec" NAME "{" meta_block documentation_block? data_block
transformed_data_block? parameters_block transformed_parameters_block? model_block
inference_block? output_block? "}"

meta_block: "meta" "{" meta_pair* "}"
meta_pair: NAME ":" value ";"

documentation_block: "documentation" (MULTILINE_STRING_LITERAL | ESCAPED_STRING)

data_block: "data" "{" data_decl* "}"
data_decl: "source" NAME ":" type (constraint)? "from" source_type "(" arg_list? ")" ";"

trans_data_decl: NAME ":" type (constraint)? ";"

constraint: "where" bool_expr

?bool_expr: bool_expr "and" bool_expr  -> and_
          | bool_expr "or" bool_expr   -> or_
          | "not" bool_expr            -> not_
          | expr
          | "(" bool_expr ")"
source_type: NAME

transformed_data_block: "transformed_data" "{" (trans_data_decl | assign_stmt |
compound_assign_stmt | for_stmt | if_stmt)* "}"

parameters_block: "parameters" "{" param_decl* "}"
param_decl: NAME ":" type constraint? prior? ";"
prior: "~" distribution

transformed_parameters_block: "transformed_parameters" "{" (trans_data_decl | assign_stmt |
```

```
compound_assign_stmt | for_stmt | if_stmt)* "}"

model_block: "model" "{" stmt_body* "}"

distribution: NAME "(" [expr ("," expr)*] ")"

inference_block: "inference" "{" "method" ":" NAME ";" settings_block? "}"
settings_block: "settings" ":" "{" inference_setting* "}"
inference_setting: NAME ":" value ";"

output_block: "output" "{" output_stmt* "}"
output_stmt: NAME ":" value_or_list ";"

arg_list: arg ("," arg)*
arg: NAME "=" value

value_or_list: value | "[" [value ("," value)*] "]"
value: NUMBER | BOOLEAN | ESCAPED_STRING | NAME

for_stmt: "for" "(" NAME "in" expr ":" expr ")" "{" stmt_body* "}"
if_stmt: "if" "(" bool_expr ")" "{" then_body "}" ("else" "{" else_body "}")?
then_body: stmt_body*
else_body: stmt_body*
assign_stmt: assign_target "=" expr ";"
compound_assign_stmt: assign_target (PLUS_EQ | MINUS_EQ | STAR_EQ | DIV_EQ) expr ";"
assign_target: NAME | indexed_access

dist_stmt: assign_target "~" distribution ";"

?stmt_body: assign_stmt | compound_assign_stmt | dist_stmt | for_stmt | trans_data_decl |
if_stmt

?expr: expr GT term   -> gt
     | expr LT term   -> lt
     | expr GTE term  -> gte
     | expr LTE term  -> lte
     | expr EQ term   -> eq
     | expr NEQ term  -> neq
     | expr PLUS term -> add
     | expr MINUS term -> sub
     | term

?term: term STAR factor -> mul
     | term DIV factor  -> div
     | term POW factor  -> pow
     | factor

?factor: function_call
       | log_prob_call
       | NAME
       | indexed_access
       | NUMBER
       | ESCAPED_STRING
       | BOOLEAN
       | "(" expr ")"

function_call: NAME "(" [expr ("," expr)*] ")"
log_prob_call: NAME "(" expr ("|" expr ("," expr)*) ")"

PLUS: "+"
MINUS: "-"
STAR: "*"
DIV: "/"
POW: "^"
GT: ">"
LT: "<"
GTE: ">="
LTE: "<="
EQ: "=="
NEQ: "!="
PLUS_EQ: "+="
MINUS_EQ: "-="
STAR_EQ: "*="
DIV_EQ: "/="
```

```
BOOLEAN: "true" | "false"
NAME: /[-]?[a-zA-Z_][a-zA-Z0-9_]*/
indexed_access: NAME "[" index_list "]"
index_list: index_part ("," index_part)?
index_part: expr?
type: array_type | base_type type_suffix?
BASE_TYPE: INT | REAL | VECTOR | MATRIX | SPARSE_MATRIX | ORDERED | SIMPLEX | BOOL
base_type: BASE_TYPE
type_suffix: "[" type_size ("," type_size)* "]"
type_size: expr
array_type: "array" "[" expr ("," expr)* "]" type
NUMBER: /-?[0-9]+(\.[0-9]+)?/

INT: "int"
REAL: "real"
VECTOR: "vector"
MATRIX: "matrix"
SPARSE_MATRIX: "sparse_matrix"
ORDERED: "ordered"
SIMPLEX: "simplex"
BOOL: "bool"

%import common.ESCAPED_STRING
%import common.WS
%ignore WS
MULTILINE_STRING_LITERAL: /"""(?:[^"\\]|\\.|""(?!"))*"""/

LPAR: "("
RPAR: ")"
```

## B.3  EXAMPLE: LINEAR REGRESSION IN STATMODELDSL

The following example illustrates a simple linear regression task defined using our DSL:

### Example: Simple Linear Regression in StatModelDSL

```
stat_model_spec linear_regression_example {
  meta {
    author: "StatBot";
    dsl_version: "1.0";
    model_version: "1.0";
    target_language: "stan";
  }

  documentation """
  Simple linear regression: predict y using x.
  """

  data {
    source N: int from csv(path="data.csv", head="num");
    source x: vector from csv(path="data.csv");
    source y: vector[N] from csv(path="data.csv");
  }

  transformed_data {
    x_centered: vector[N];
    x_mean: real;
    x_mean = mean(x);
    for (i in 1:N) {
      x_centered[i] = x[i] - x_mean;
    }
  }

  parameters {
    alpha: real ~ normal(0, 10);
    beta: real ~ normal(0, 5);
    sigma: real where sigma > 0 ~ exponential(1);
  }

  model {
    for (i in 1:N) {
      y[i] ~ normal(alpha + beta * x_centered[i], sigma);
    }
  }
```

```
inference {
  method: nuts;
  settings: {
    chains: 4;
    num_samples: 1000;
    num_warmup: 500;
  }
}

output {
  monitor: [alpha, beta, sigma];
  summary_stats: [mean, q5, q95];
  diagnostics: [rhat];
  plots: [alpha, beta];
  export_results_to: "results_linear.csv";
}
}
```

If we set "target_language" to "stan", the transformed stan code is:

**Stan code for model inference**

```
data {
  int N;
  vector[100] x;
  vector[N] y;
}
transformed data {
  vector[N] x_centered;
  real x_mean = mean(x);
  x_mean = mean(x);
  for (i in 1:N) {
    x_centered[i] = x[i] - x_mean;
  }
}
parameters {
  real alpha;
  real beta;
  real<lower=0.0> sigma;
}
model {
  alpha ~ normal(0, 10);
  beta ~ normal(0, 5);
  sigma ~ exponential(1);
  for (i in 1:N) {
    y[i] ~ normal(alpha + beta * x_centered[i], sigma);
  }
}
```

If we set "target_language" to "pymc", the transformed PyMC code is:

**PyMC code for model inference**

```
def build_and_sample(data, random_seed=None,
return_inference_data=True, **sample_kwargs):
    import pymc as pm
    import numpy as np
    import aesara.tensor as at
    import arviz as az
    model = pm.Model()
    with model:
        N = pm.MutableData('N', data.get('N'))
        x = pm.MutableData('x', data.get('x'))
        y = pm.MutableData('y', data.get('y'))
        x_centered = x - at.mean(x)
        alpha = pm.Normal('alpha', 0, 10)
        beta = pm.Normal('beta', 0, 5)
        sigma = pm.Exponential('sigma', 1)
        mu = alpha + beta * x_centered
        pm.Normal('y', mu=mu, sigma=sigma, observed=y)
        chains = sample_kwargs.pop('chains', 4)
        draws = sample_kwargs.pop('draws', 1000)
        tune = sample_kwargs.pop('tune', 500)
        target_accept = sample_kwargs.pop('target_accept', 0.8)
```

```python
        trace = pm.sample(draws=draws, tune=tune, chains=chains,
        target_accept=target_accept, random_seed=random_seed, ** sample_kwargs)
        if return_inference_data:
            return az.from_pymc3(trace)
        return trace
```

## C    STATMODELCHATBOT

### C.1    EXTRACTION SCHEMA

We design an extraction schema to facilitate the StatModelChatbot to verify all necessary task information. Here is the detailed schema we design:

```
{
    "data": {
        "variable_name": {
            "variable_type": "",
            "data_path": ""
        },
        ...
    },
    "transformed_data(optional)": {
        "variable_name": {
            "variable_type": "",
            "expression": ""
        },
        ...
    },
    "parameters": {
        "variable_name": {
            "variable_type": "",
            "distribution": "",
            "constraint(optional)": ""
        },
        ...
    },
    "transformed_parameters(optional)": {
        "variable_name": {
            "variable_type": "",
            "expression": ""
        },
        ...
    },
    "model": [
        "expression1", "expression2", ...
    ],
    "inference": [
        "configuration1", "configuration2", ...
    ],
    "output": [
        "configuration1", "configuration2", ...
    ]
}
```

### C.2    STATMODELCHATBOT WORKFLOW

As illustrated in Figure 2 (a), our StatModelChatbot goes through five sequential nodes to complete the entire schema and output a final task description. Here are the detailed prompts for our conversational agent.

**Prompt for task definition**

```
You are given a user's description of a dataset for a statistical modeling task. Please use
a short paragraph to summarize what this task intends to do.

Ouput format:
```markdown
<your answer>
```

```
```

---

User:
{user}

Example output:
{example}
```

**Prompt for data loading**

```
You are an assistant who helps structure user descriptions into  a predefined schema.
The current task is about the **input data (data node)**.

The schema is defined as follows:
{
    "data": {
        "variable_name": {
            "variable_type": "",
            "data_path": ""
        },
        ...
    }
}
'''
    prompt = prompt + f'''

Instructions:
1. Carefully read the user's description:
   {user}

2. Fill in the schema above with as much information as possible.
   - `variable_name`: the name of the variable or dataset.
   - `variable_type`: the type of the variable (e.g., int, real, vector, array).
   - `data_path`: the exact path or identifier of the data
   (e.g., `benchmark/data/data1.csv`).

3. If the description provides enough information to complete a field, fill it in.
   If some required fields are missing, leave them as empty strings `""`.

4. Provide **two outputs**:
   - **Schema output** (inside ```json ... ```).
   - **Feedback in natural language** (inside ```markdown ```).
     - If all required information is present, the first line of Feedback must be `Enough`.
     - If some required information is missing, the first line must be `Not Enough`,
     followed by an explanation of what is missing and what the user should provide.

Make sure to always output both parts.
---
Examples:
{example}
```

**Prompt for variable specification**

```
You are an assistant that helps structure user descriptions into a predefined schema.
The current task is about the **variable node**.

The schema is defined as follows:
{
    "transformed_data(optional)": {
        "variable_name": {
            "variable_type": "",
            "expression": ""
        },
        ...
    },
    "parameters": {
        "variable_name": {
            "variable_type": "",
            "distribution": "", "
            constraint(optional)": ""
        },
        ...
```

```
        },
        "transformed_parameters(optional)": {
            "variable_name": {
                "variable_type": "",
                "expression": ""
            },
            ...
        }
}

Instructions:
1. Carefully read the user's description:
    {user}

2. Fill in the schema above with as much information as possible.
    - `parameters` is **required**. Each parameter should have:
        - `variable_name`: the name of the parameter.
        - `variable_type`: the type (e.g., int, real, vector, array).
        - `distribution`: the assumed prior distribution.
        - `constraint(optional)`: optional constraints (e.g., >0,
        between 0 and 1).
    - `transformed_data` and `transformed_parameters` are **optional**.
        - If mentioned in the description, fill them in.
        - If not mentioned, you may leave them out without asking the user.

3. If the description provides enough information to fully specify the required fields in
`parameters`, fill them in.
    If some required fields are missing, leave them as empty strings `""`.

4. Provide **two outputs**: (the same as data loading)

Make sure to always output both parts.
---
Examples:
{example}
```

---

## Prompt for model definition

```
You are an assistant that helps structure user descriptions into a predefined schema.
The current task is about the **model node**.

The schema is defined as follows:
{
    "model": [
        "expression1", "expression2", ...
    ]
}

Instructions:
1. Carefully read the user's description:
    {user}

2. Extract the part that describes the model (if any).
    - If the user mentions a model structure, equations, or likelihoods, summarize them as a
    list of expressions.
    - If the user does not provide any model information, leave the list empty.

3. Provide **two outputs**:
    - **Schema output** (inside ```json ... ```).
    - **Feedback in natural language** (inside ```markdown ... ```).
        - If model expressions are found, the first line of Feedback must be `Enough`, followed
        by a short summary.
        - If no model information is found, the first line must be `Not Enough`, followed by a
        clear request for the user to provide model details.

Make sure to always output both parts.
---
Examples:
{example}
```

```
Prompt for result configuration

You are an assistant that helps structure user descriptions into a predefined schema.
The current task is about the **inference and output nodes**.

The schema is defined as follows:
{
    "inference": [
        "configuration1", "configuration2", ...
    ],
    "output": [
        "configuration1", "configuration2", ...
    ]
}

Instructions:
1. Carefully read the user's description:
    {user}

2. Identify whether the description includes any details about:
    - **Inference** (e.g., sampling method, number of iterations, variational inference,
    optimization settings, etc.).
    - **Output** (e.g., what results to return, summary statistics, predictions, posterior
    samples, etc.).

3. Provide **two outputs**:
    - **Schema output** (inside ```json ... ```), filling in the configurations if available.
    - **Feedback in natural language** (inside ```markdown ... ```):
      - If inference and/or output details are found or the user says just default settings,
      the first line must be `Enough`, followed by a short summary.
      - If no inference/output information is found, the first line must be `Not Enough`,
      followed by a clear request for the user to specify inference and/or output
      configurations.

Make sure to always output both parts.

---
Examples:
{example}
```

# D    AN ENTIRE EXAMPLE USING AUTOSTAT, FROM TASK SPECIFICATION TO DSL

First, the user can chat with the StatModelChatbot to specify the entire task, illustrated in Table 5.

The StatModelChatbot will return the final output including the schema and the task description:

```
Output schema

{
    "data": {
        "N": {
            "variable_type": "int",
            "data_path": "testdata.json"
        },
        "year": {
            "variable_type": "vector[int][N]",
            "data_path": "testdata.json"
        },
        "consumption": {
            "variable_type": "vector[N]",
            "data_path": "testdata.json"
        },
    },
    "transformed_data": {
        "diff_consumption": {
            "variable_type": "vector[N-1]",
            "expression": "diff_consumption[i] = consumption[i+1] - consumption[i]
for i in 1 to N-1"
        }
    },
    "parameters": {
```

Table 5: An example of the dialogue between user and StatModelChatbot.

| User | We're modeling year-to-year changes in electricity consumption using a simple Bayesian regression. After computing the first-order differences of consumption, we regress these differences on the corresponding years with a normal likelihood. |
|------|------|
| Chatbot | Please provide the data paths for the electricity consumption differences and corresponding years data files. |
| User | Our data is testdata.json, with an interger N and two vectors year and consumption. |
| Chatbot | The description mentions using a normal likelihood for regression but doesn't specify the prior distributions for alpha, beta, and sigma. Please provide the prior distributions for these parameters (e.g., normal priors for alpha and beta, half-normal or half-cauchy for sigma). |
| User | alpha is Normal(0, 10), beta Normal(0, 5), sigma is exp(1) with lower bound 0 |
| Chatbot | The description provides details about the Bayesian regression model structure (priors for alpha, beta, sigma, and likelihood) and the data source, but it doesn't specify any inference configurations (sampling method, number of iterations, chains, etc.) or output configurations (what results to return, summary statistics, predictions, or where to save outputs). Please specify your inference and output preferences, or indicate if you'd like to use default settings. |
| User | For inference, I think just take default settings is ok. Use nuts sampling method, with 4 chains. I need 1000 warmup steps, then 1000 steps. For output, I need to see the median, ess, plots for alpha and beta. Remember to export results to this file: electricity_consumption_results.csv |

```
          "alpha": {
              "variable_type": "real",
              "distribution": "normal(0, 10)"
          },
          "beta": {
              "variable_type": "real",
              "distribution": "normal(0, 5)"
          },
          "sigma": {
              "variable_type": "real",
              "distribution": "exponential(1)",
              "constraint": "> 0"
          }
      },
      "model": ["diff_consumption[i] ~ normal(alpha + beta * year[i+1], sigma)
  for i in 1:(N-1)"
      ],
      "inference": [
          "Use `nuts` method",
          "4 chains",
          "1000 samples after 1000 warmup iterations"
      ],
      "output": [
          "Report median as summary statistic",
          "Check effective sample size (ess) for diagnostics",
          "Generate plots for `alpha` and `beta`",
          "Export results to `electricity_consumption_results.csv`"
      ]
  }
```

## Output task definition

```
You are provided with a dataset at `testdata.json`, which includes:
- `N`: an integer representing the number of observations.
- `year`: a vector of length `N` containing the observation years.
- `consumption`: a vector of length `N` with consumption values (constrained to be greater
than 0).

Additionally, you define a transformed variable:
- `diff_consumption`: a vector of length `N-1`, calculated as
`consumption[i] - consumption[i-1]` for each `i` from 2 to N.

You defined all required parameters clearly:
- `alpha ~ normal(0, 10)`
- `beta ~ normal(0, 5)`
- `sigma > 0 ~ exponential(1)`

You also introduced a transformed data variable:
- `diff_consumption`: a vector of size `N-1`, where each element is
`consumption[i+1] - consumption[i]`.

Model specification:
- For each `i` in `1:(N-1)`, the differences in consumption are modeled as
  `diff_consumption[i] ~ normal(alpha + beta * year[i+1], sigma)`.
This means consumption changes are assumed to follow a normal distribution with mean
determined by a linear function of the year and standard deviation `sigma`.

Inference configuration:
- Use the `nuts` sampler with 4 chains.
- Run 1000 warmup iterations followed by 1000 samples.

Output configuration:
- Report the **median** as the main summary statistic.
- Check **effective sample size (ess)** for diagnostics.
- Generate **plots for alpha and beta**.
- Export results to **`electricity_consumption_results.csv`**.
```

Finally, our StatModelCopilot will generate the corresponding DSL code as follows:

**Output task definition**

```
stat_model_spec electricity_consumption_model {
  meta {
    author: "StatBot";
    dsl_version: "1.0";
    model_version: "1.0";
    target_language: "stan";
  }

  documentation """
  Bayesian regression model for year-to-year changes in electricity consumption.
  Models first-order differences of consumption regressed on years with normal likelihood.
  """

  data {
    source N: int from json(path="testdata.json");
    source year: vector[N] from json(path="testdata.json");
    source consumption: vector[N] from json(path="testdata.json");
  }

  transformed_data {
    consumption_diff: vector[N-1];
    year_diff: vector[N-1];

    for (i in 1:(N-1)) {
      consumption_diff[i] = consumption[i+1] - consumption[i];
      year_diff[i] = year[i+1];
    }
  }

  parameters {
    alpha: real ~ normal(0, 10);
    beta: real ~ normal(0, 5);
    sigma: real where sigma > 0 ~ exponential(1);
  }

  model {
    for (i in 1:(N-1)) {
      consumption_diff[i] ~ normal(alpha + beta * year_diff[i], sigma);
    }
  }

  inference {
    method: nuts;
    settings: {
      chains: 4;
      num_samples: 2000;
      num_warmup: 1000;
    }
  }

  output {
    monitor: [alpha, beta, sigma];
    summary_stats: [median, ess];
    diagnostics: [rhat];
    plots: [alpha, beta];
    export_results_to: "electricity_consumption_results.csv";
  }
}
```

# E  STATMODELDATASET

In this section, we will deliver more details about our dataset construction process, including prompt for data generation and prompt for data augmentation.

**Prompt for data generation**

```
You are a helpful DSL generator. I will provide you with:
- The grammar of the StatModelDSL,
- A few example DSL programs,
- A target usage scenario,
- And a set of configuration settings.
```

Your task is to generate a complete StatModelDSL program that fits the given scenario and
settings. Strictly follow the provided grammar and take inspiration from the examples.
Additionally, create a detailed task description written from the user's perspective -
imagine it's how someone would explain their modeling needs to an assistant.

You are allowed to define any `.csv` or `.json` data sources in your DSL program, but you
do **not** need to generate their contents - we will provide the actual data files.

Please output your response in **exactly** the following format:

Description:

```markdown
<The user's prompt when he want copilot to generate this
program. 1-3 sentences is ok.>
```

DSL code:

```dsl
<your complete DSL program here>
```

---

**Inputs:**

DSL Grammar:
```markdown
{grammar}
```

DSL Program Examples:
```dsl
{examples}
```

Usage Scenario:
In the domain of **{domain}**, the task is **{task}**

---

## Prompt for data augmentation

You are a helpful assistant for StatModelDSL, a DSL similar to Stan. I will give you:

* A StatModelDSL code snippet (which will be compiled into Stan).
* A natural-language task description written by a user.
* A few task examples demonstrating how to rewrite the natural-language description to
fully reflect the DSL specification.

Your job is to **rewrite the user's task description** so that it precisely
corresponds to the provided DSL code. The rewritten description should specify:

1. All data inputs and their types (e.g., int, real, vector, array), and where they
come from.
2. Any derived (transformed) data or parameters and how they are computed.
3. All model parameters and their prior distributions (with parameter values).
4. The structure of the model (e.g., likelihoods, regression equations, etc.).
5. Inference settings such as number of chains, number of samples, warm-up iterations.

### Style instructions:

* The final description should **sound like a power user** of the tool giving precise
instructions to the system, written in a natural, fluent tone, as if to a copilot.
* Keep it concise but complete. Avoid vague language.
* Don't explain what the DSL does - describe *what the user wants* in a way that fully
specifies the model.

Please output your response using this format:

---

Task description:

```markdown
<your rewritten description here>
```

```
```

---

Input:
Simple task description:

```markdown
{description}
```

DSL code:

```dsl
{dsl}
```

Examples:
{examples}
```

## F  STATMODELCOPILOT

We choose Llama3.1-8B-Instruct (Dubey et al., 2024) as our base LLM to train our StatModelCopilot. The implementation details are shown in Table 6. All experiments are conducted on a single NVIDIA A40 GPU, supported by Llama-Factory (Zheng et al., 2024).

Table 6: Implementation details and training cost.

| Stage | Learning Rate | Lora Rank | # Training Data | # Training Epochs | Training Time |
|-------|---------------|-----------|-----------------|-------------------|---------------|
| **One** | $1 \times 10^{-4}$ | 32 | 5064 | 5 | 3h |
| **Two** | $5 \times 10^{-5}$ | 32 | 10907 | 3 | 7h |

## G  MORE EXPERIMENTAL DETAILS

### G.1  DETAILS ABOUT OUR TEST DATASET

Table 7: Details about our test set. All "lengths" mean the average lengths of all data. "Entire" means the entire test dataset.

|  | **Simple** | **Medium** | **Complex** | **Entire** |
|--|-----------|-----------|------------|-----------|
| # Data | 81 | 165 | 77 | 323 |
| Insrtuction Length $X_d$ | 1676 | 1917 | 2153 | 1913 |
| DSL code length $Y$ | 1091 | 1331 | 1615 | 1339 |

Table 7 demonstrates more details about our test dataset. We decompose our test set into 3 levels based on the length of the instructions to evaluate how different input lengths will affect the performance.

### G.2  LLM-AS-A-JUDGE

For quantitative experiments, we leverage GPT-4o as a judge to list out the mismatching items between the description and the generated DSL/PPL code. Here is the prompt:

```
I will provide you with:
- A DSL code program
- The corresponding detailed description of the DSL code about the statistical modeling task

Please help me to check if the description exactly matches the code. You need to focus on
each line of the description! Please list all the mismatches one by one.

If match, just answer: Match
If not, answer in this format:
Not match.
1. In description, we need xxx, but in the code, xxx
2. xxx

---

DSL code:
```dsl
{code}
```

Description:
```markdown
{description}
```
```

### G.3    User Study

**Experimental settings.**    Each participant is asked to perform a simple code modification task in
each of the different programming environments. The purpose is to give them a basic familiarity with
the environment and ensure that they have carefully reviewed the provided code. After completing
the tasks in all three environments, participants are asked to rank the code in terms of clarity and
readability, and then indicate their preference—that is, which environment they would choose when
encountering a similar task in the future. To ensure the validity of the results and avoid any order
effects, the sequence in which participants worked with the three environments was randomized. The
tasks they encountered were also assigned randomly, while maintaining similar levels of difficulty
across conditions.

**Participants.**    We collected a total of 17 responses. None of the participants had prior experience
with PPLs such as Stan or PyMC, but all had a solid foundation in Python and data analysis, as well
as a basic understanding of statistical concepts. From their responses, we can gain valuable insights
into novice users' preferences and perceptions of different programming environments.

**Results Analysis.**    From the results shown in Figure 4, we can observe that: 1) Clarity: Over half
of the participants favored our DSL, demonstrating that its design—by omitting verbose PPL code
and general-purpose language code (here, Python)—is concise and highly readable. 2) Preference:
Our DSL also achieved relatively high preference scores, indicating that for novices, this DSL for-
mat is indeed easier to use. However, compared with Stan, the advantage is not particularly large.
We believe this is because participants were already very familiar with Python, and the statistical
components in Stan were relatively minimal and straightforward. Additionally, the Stan code itself
is fairly concise and easy to understand, which led many users to also find it accessible.

## H    Real-world Evaluation

To further demonstrate the practicality and effectiveness of our StatModelDSL and the full AutoStat
framework in real-world scenarios, we conduct a case study on two representative settings: 1) repro-
ducing real examples from the Stan textbook, and 2) replicating experiments reported in published
statistical research papers.

### H.1    Task Collections

We collect real-world Bayesian modeling tasks from two types of scenarios:

- **Textbook examples**. We collect real-world Stan-based code from the official Stan example repository[3]. These examples are drawn from three textbooks Gelman & Hill (2007); Lee & Wagenmakers (2014); Kéry & Schaub (2011). We first filter out overly simplistic or incomplete code, then randomly select 50 tasks from the remaining set. Each task is then converted into a complete natural language description using GPT-4o (Hurst et al., 2024), and subsequently implemented in StatModelDSL via our AutoStat framework.

- **Research paper experiments**. We collect all papers published in **Bayesian Analysis**[4] from 2022 onward. For each paper, we extracted all simulation experiments that either relied on Bayesian modeling or could be implemented using Bayesian approaches as a baseline. Using GPT-4o (Hurst et al., 2024), we converted these experiments into structured task descriptions, filtering out cases with overly vague descriptions or missing experimental details. From the remaining set, we randomly selected 50 tasks, which were implemented and simulated using StatModelDSL through our AutoStat framework. Because some experimental descriptions in the source papers are not sufficiently detailed and the resulting outputs span diverse dimensions, it is difficult to design rule-based checks to validate correctness. Therefore, our evaluation relies on a combination of syntactic validation, manual inspection, and LLM-as-a-judge assessment to determine whether the generated DSL faithfully matches the task requirements.

**All generated DSL programs are syntactically correct**, and the full set of programs is available in our official code repository.

## H.2 EXAMPLES

For textbook examples, here is a case:

---

**Task definition**

```
Develop a Bayesian hierarchical model using Stan to analyze data from multiple counties,
focusing on estimating county-specific effects on a continuous outcome variable. The
model includes hierarchical priors for county effects and uses NUTS sampling for inference,
with key parameters including the overall mean, county-specific deviations, and their
variances. The output should include plots of the primary parameters to assess their
distributions and relationships.

(More details omitted)
```

---

**DSL implementation**

```
stat_model_spec hierarchical_county_model {
  meta {
    author: "Public Health Analyst";
    dsl_version: "1.0";
    model_version: "1.0";
    target_language: "stan";
  }

  documentation """
  Bayesian hierarchical model for county-level data, using non-centered parameterization.
  """

  data {
    source J: int from json(path="testdata.json");
    source N: int from json(path="testdata.json");
    source county: array[N] int from json(path="testdata.json");
    source y: vector[N] from json(path="testdata.json");
  }

  parameters {
    eta: vector[J] ~ normal(0, sigma_eta);
    mu: real ~ normal(0, 1);
    sigma_eta: real where sigma_eta > 0 ~ uniform(0, 100);
    sigma_y: real where sigma_y > 0 ~ uniform(0, 100);
  }
```

---

[3]https://github.com/stan-dev/example-models
[4]https://projecteuclid.org/journals/bayesian-analysis

```
    transformed_parameters {
      eta_adj: vector[J];
      mean_eta: real;
      mu_adj: real;
      for (j in 1:J) {
        eta_adj[j] = eta[j] - mean(eta);
        mean_eta = mean(eta);
      }
      mu_adj = 100 * mu + mean_eta;
      for (i in 1:N) {
        y_hat: vector[N];
        for (j in 1:J) {
          y_hat[i] = 100 * mu + eta[county[i]];
        }
      }
    }

    model {
      mu ~ normal(0, 1);
      sigma_eta ~ uniform(0, 100);
      sigma_y ~ uniform(0, 100);
      for (j in 1:J) {
        eta[j] ~ normal(0, sigma_eta);
      }
      for (i in 1:N) {
        y[i] ~ normal(y_hat[i], sigma_y);
      }
    }

    inference {
      method: nuts;
      settings: {
        chains: 4;
        num_samples: 1000;
        num_warmup: 500;
      }
    }

    output {
      monitor: [mu, sigma_eta, sigma_y, eta];
      summary_stats: [mean, q5, q95];
      diagnostics: [rhat];
      plots: [mu, sigma_eta, sigma_y];
      export_results_to: "results_hierarchical_county.csv";
    }
  }
```

This dsl implementation perfectly aligns with the task description and can be converted to the ground-truth stan code.

For research paper examples, here is a case: In Section S4 of the appendix in Beraha et al. (2024), the authors conducted a Bayesian analysis on a mixture of Gaussians to compare the performance of different sampling methods. Here, we replicate this experiment using StatModelDSL, performing MCMC sampling with Stan. The corresponding DSL code is as follows:

```
stat_model_spec mixture_gaussian {
  meta {
    author: "User";
    dsl_version: "1.0";
    model_version: "1.0";
    target_language: "stan";
  }

  documentation """
  We want to build a mixture-of-Gaussian distribution and test
  the inference success rate and time cost of MCMC sampling.
  """

  data {
    source N: int where N > 1 from json(path="testdata.json");
    source K: int where K > 1 from json(path="testdata.json");
    source y: vector[N] from json(path="testdata.json");
  }
```

```
    parameters {
      theta: simplex[K] ~ dirichlet(rep_vector(1.0, K));
      sigma: vector[K] ~ cauchy(0, 2.5);
      mu: vector[K] ~ normal(0, 5);
    }

    model {
      // Likelihood: mixture density
      for (n in 1:N) {
        lps: vector[K];
        for (k in 1:K) {
          lps[k] = log(theta[k]) +
              normal_lpdf(y[n] | mu[k], sigma[k]);
        }
        target += log_sum_exp(lps);  // mixture log-likelihood
      }
    }

    inference {
      method: nuts;
      settings: {
        chains: 4;
        num_samples: 2000;
        num_warmup: 1000;
      }
    }

    output {
      monitor: [theta];
      summary_stats: [mean, q5, q95];
      diagnostics: [rhat, ess];
      export_results_to: "results.csv";
      posterior_predictive_checks: true;
      age_group_differences: true;
      credible_intervals: [0.9, 0.95];
    }
  }
```

Our experimental results show that when $n = 100$, the effective sample size (ESS) is 22.33 with a sampling time of 2.58s; when $n = 250$, the ESS decreases to 14.61 with a sampling time of 3.78s. Compared with Figure 1 in Section S4 (Beraha et al., 2024), these results fall within a consistent and realistic range.

