# OpenReview forum: "AutoStat: DSL-based Automated Statistical Modeling from Natural Language"
_ICLR.cc/2026/Conference — Submitted to ICLR 2026_

### Official Review · Reviewer_JFrL · 2025-10-17

**Soundness:** 2
**Presentation:** 3
**Contribution:** 2
**Rating:** 2
**Confidence:** 3

**Summary:**

The presents a domain specific language framework, called AutoStat, to automate statistical modelling. The framework consists of:
- A novel DSL named StatModelDSL, which can compile to multiple PPL frameworks, specifically listed are Stan and PyMC3
- An agent/fine-tuned LLM, StatModelChatbot, which refines user queries
- An agent/fine-tuned LLM, StatModelCopilot, which generates programs in StatModelDSL from the user specifications
Empirically, the paper evaluates AutoStat framework on a test set of synthetically generated modelling problems. AutoStat is able to provide higher accuracy in generating syntactically valid DSL code, as well as, providing output programs that more closely match the specified target program, as evaluated by an LLM judge.

**Strengths:**

The task of improving how non-experts can leverage PPLs for improved statistical modelling is definitely valuable. Furthermore, the authors seem to clearly but a lot of work into the non-trivial task of generating a dataset for statistical modelling task, even though it is fully synthetic (a downside discussed in the weaknesses).

In general, the paper is easy to read and the main ideas can be followed easily.

Overall, I am skeptical though of the benefit of using a purpose-built DSL for LLMs for this use case which I discuss in the weaknesses below.

**Weaknesses:**

Multiple high-level weaknesses of the paper currently make me vote for rejection:

1. Overall, I don't see a need for designing a novel DSL that is mainly designed for the usage by LLMs. The theoretical arguments and the experimental results provided in the paper do not convince me otherwise. I base this judgement on the following three points:
  - The paper says the proposed DSL offers "Completeness" (L.70) of being able to represent the full data analysis pipeline necessary for statistical modelling. But there is no proof  (either theoretical or empirical) provided in the paper that the DSL is actually in any sense of the word "complete" and the experimental evaluation is exclusively done on synthetic data. Fundamentally, any DSL will have edge cases that cannot be represented with it. I think this downside of choosing to formulate a new DSL is not discussed enough in the paper.
  - The DSL supposedly provides "Portability" (L.71) because it can compile to Stan and PyMC but in practice there will only be a limited number of PPLs that the DSL can compile into. Effectively, there is still a lock-in into the PPLs that AutoStat has as compilation targets.
  - LLMs are already improving rapidly on coding tasks, making it unclear whether developing a DSL specifically for LLMs and statistical modelling is necessary. This is backed up by Table 1. GPT-4o with in-context learning outperforms both the AutoStat-1B and AutoStat-3B models. I would assume that more recent GPT or Claude models would have performance that is competitive with AutoStat-8B.

2. The paper presents a framework for automated statistical modelling *but the generated statistical models are never evaluated based on their statistical properties*. None of the following questions below are answered:
- What is the test set/cross-validation performance of the generated statistical models?
- How easy is statistical inference in the generated models (e.g. number of divergences in HMC samples, effective sample size (ESS), etc.)?
- What is the performance of the whole set-up on real-world data and modelling tasks? As far as I can tell, the evaluation is limited to synthetically generated problems.

3. One of the headline metrics to praise the performance of AutoStat is that it achieves "100 % syntax correctness rate for DSL generation" (L28). But this does not seem to be a very difficult goal to achieve? Tools such as outlines can guaruantee that you will generate syntactically valid programs with arbitrary base LLMs.

**Questions:**

There are very few details on the conducted user study in the main text and the appendix. Additionally,  only 17 participants for a user study seems to be quite small to have conclusive results and there is not much detail on the study design and what was done to ensure the results are statistically significant. Additionally, it would be good to have more (anonymised) details about the study participants, are these professional software developers or undergraduates, etc.?

Minors:
L. 129: Missing whitespace in front of citation.

---

> ### Author Response · Authors · 2025-11-17
>
> Your feedback is very insightful, and the questions you raised are crucial. We will respond to your weaknesses and questions one by one.
>
> **W1: Why we design our DSL**
>
> **1. The DSL is not designed to replace PPLs, but to serve as a structured intermediate representation.**
>
> Our goal is not to argue that existing statistical modeling workflows or PPLs are poorly designed. Instead, our focus is on simplifying the end-to-end process when the task begins with a natural-language description. The proposed DSL is not intended to replace PPLs; rather, it serves as a structured intermediate representation that bridges natural-language task descriptions and the diverse tools required for a complete modeling workflow to address three core limitations:
>
> **(a) Intrinsic complexity of statistical modeling (task-level challenge)**
>
> Real Bayesian models involve many subtle yet crucial design choices. Users and LLMs frequently omit or mis-specify these details when describing models in natural language. StatModelChatbot + StatModelDSL ensure these components are explicitly specified before executing the workflow.
>
> **(b) Fragmentation of end-to-end statistical modeling workflows (workflow-level challenge)**
>
> A full statistical modeling workflow typically spans multiple disconnected environments
>
> Current PPLs only specify the model itself; all remaining steps must be implemented separately in external scripts.
>
> StatModelDSL unifies these components within a single specification, which is then compiled into:
> + Python code for data preprocessing and output handling
> + PPL code for model specification and inference
> + a fully executable end-to-end pipeline
>
> This removes the fragmentation inherent in current workflows.
>
> **(c) Lack of interoperability across PPLs (backend-level challenge)**
>
> Different PPLs use fundamentally incompatible syntactic conventions and execution backends. As a result, models written for one PPL cannot be executed in another.
>
> To our knowledge, no tool can translate a non-trivial model between these frameworks, due to differences in syntax, semantics, and computational backends. StatModelDSL provides a backend-agnostic intermediate representation. This unifies heterogeneous PPL ecosystems under a single modeling interface.
>
> For more analysis and examples, due to space limitations, please refer to point three of the responses to reviewer `DPSW`.
>
> **2. On completeness and portability**
>
> To address completeness, we focus on practical completeness, defined as the ability to represent all major Bayesian model families commonly expressible in mainstream PPLs.
> Our DSL is designed so that the model block mirrors Stan’s expressive constructs—supporting standard distributions, constraints, transformed parameters, hierarchical structure, mixture components, and user-defined likelihood formulations.
> **Under this assumption, almost any model expressible in Stan using standard modeling patterns can also be represented in StatModelDSL**.
>
> Furthermore, in our real-world evaluation in Sec. 4.6, the DSL successfully expresses more than 100 real Bayesian models taken from Stan’s official examples and recent scientific publications. This provides strong evidence that the DSL covers the modeling patterns required in practice.
>
> The same applies to portability: we currently support multiple PPL backends (stan and PyMC, which are the most popular ones) and will continue extending this layer.
>
> **3. Why do we fine-tune our StatModelCopilot**
>
> We fully agree with your point—state-of-the-art proprietary models can indeed perform extremely well. However, training our own StatModelCopilot serves two important purposes.
> + First, it provides greater reliability and consistency for our pipeline.
> Our model achieves 100% syntactic correctness and nearly 99% semantic correctness, offering more stable behavior than open-source base models in the statistical-modeling setting.
> + Second, a self-hosted model ensures privacy, security, and deployment flexibility, which is often required in real-world applications where closed-source APIs cannot be used.

---

> ### Author Response · Authors · 2025-11-17
>
> **W2: evaluation**
>
> **1. AutoStat does not introduce a new inference algorithm.**
>
> The statistical properties of posterior samples (ESS, divergences, predictive accuracy, etc.) are determined entirely by the chosen PPL backend (Stan/PyMC/NumPyro). Since AutoStat compiles the DSL into standard PPL code, the inference behavior is identical to running the same model handwritten in that PPL. It is not our contribution at all.
>
> Instead, our contribution lies in automating model specification, not modifying inference. As shown in Sec. 4.5, our approach significantly reduces errors (92% fewer misalignments) and improves usability (41% user preference).
>
> **2. Real-world tasks have been added.**
>
> To further illustrate that AutoStat is effective in real-world scenarios, we conducted an additional real-world evaluation, using tasks collected from:
> + widely used statistical modeling textbooks [1, 2, 3], and
> + recent top-tier journal articles [4] containing complex Bayesian models.
>
> As detailed in Section 4.6 (we have updated our paper), AutoStat successfully handled these expert-crafted tasks, showing strong robustness beyond synthetic or LLM-generated descriptions.
>
> **W3: syntax passing rate**
>
> Our point is simply that 100% DSL syntax correctness ensures the stability of the overall NL → DSL → Python + PPL pipeline, since any syntax error would stop downstream compilation. More importantly, beyond syntax, our results in Table 1 (semantics), Figure 4, and Table 5 show that AutoStat produces substantially more accurate and semantically correct models, which is the core evidence of the system’s effectiveness.
>
> **Q: details about our user-study participants.**
>
> Our 17 participants include undergraduate, master’s, and PhD students in AI/statistics, most without prior statistical programming experience. **This is the first time for all participants to use our StatModelDSL**.
>
> Each participant completed 4 tasks (selected randomly from a 10-task pool), including code editing and representation-preference ranking, which reduces ordering bias. A sample size of 12–20 participants is standard for formative usability studies [5], and our goal is to evaluate usability trends rather than perform statistical hypothesis testing.
>
> We will add anonymized demographics and details of the study design.
>
> We hope these clarifications and our updated manuscript fully address your concerns, and we sincerely appreciate your constructive feedback. Please feel free to raise any additional questions.
>
> [1] Gelman, Andrew, and Jennifer Hill. Data analysis using regression and multilevel/hierarchical models. Cambridge university press, 2007.
>
> [2] Michael D Lee and Eric-Jan Wagenmakers. Bayesian cognitive modeling: A practical course. Cambridge university press, 2014.
>
> [3] Marc K´ery and Michael Schaub. Bayesian population analysis using WinBUGS: a hierarchical perspective. Academic press, 2011.
>
> [4] Bayesian Analysis https://projecteuclid.org/journals/bayesian-analysis
>
> [5] Nielsen, Jakob, and Thomas K. Landauer. "A mathematical model of the finding of usability problems." Proceedings of the INTERACT'93 and CHI'93 conference on Human factors in computing systems. 1993.

---

> > ### Comment · Reviewer_JFrL · 2025-11-24
> >
> > Thank you for the extensive response.
> >
> > I appreciate the addition of the new real-world modeling task.
> >
> > I still have serious concerns about the evaluation of the generated models in the paper though. The fact that "AutoStat does not introduce a new inference algorithm. [...]" does not mean that AutoStat generated programs do not need to be evaluated on their inference properties. Especially for inference algorithms like HMC, the inference algorithm performance can be quite sensitive to how the model is defined (see e.g. https://mc-stan.org/docs/2_18/stan-users-guide/reparameterization-section.html for a classic example from the Stan documentation).
> >
> > Similarly, a metric like predictive performance on a held out test set, or some other version of cross-validation, is a much simpler and objective way to evaluate the quality of generated models than having the generated models be assessed by LLMs.
> >
> > In general, I think the work has promise to be published with some significant revisions (including the ones other reviewers highlighted) but I am afraid I am maintaining my current score for now.

---

> > > ### Author Response · Authors · 2025-11-25
> > > **Supplementary for HMC inference experiments**
> > >
> > > Thank you for the suggestion!
> > >
> > > Previously, we did not include HMC inference diagnostics because Table 1 evaluates the equivalence of the generated model *specifications* (block structure, distributions, parameterization, etc.), which verifies correctness on the modeling level. However, as you point out, inference-level comparison is also important when benchmarking against actual modeling pipelines.
> > >
> > > To address this, we conducted new inference experiments based on the **real-world textbook models** used in our application study.
> > >
> > > We selected these models because the accompanying GitHub repository provides real, complex datasets that are more representative and higher quality than synthetic datasets. Due to the high computational cost of HMC on these models, we randomly sampled **10 tasks** and evaluated the following **core Stan diagnostics**, which are widely recognized as the gold standard:
> > >
> > > - **Divergences**: We treat *0 divergences* as a successful inference run.
> > > - **$\hat{R}$**: Convergence diagnostic; *$\hat{R} < 1.1$* indicates convergence.
> > > - **ESS (bulk ESS)**: Effective sample size; we consider *ESS > 100* acceptable.
> > >
> > > We compared AutoStat against the original Python + Stan implementations, using a fixed sampling configuration:
> > >
> > > - iter_warmup = 200
> > > - iter_sampling = 1000
> > > - chains = 4
> > > - adapt_delta = 0.8
> > >
> > > **Summary of the results:**
> > >
> > > | Method        | Divergences (= 0) | $\hat{R} < 1.1$ | ESS > 100 |
> > > | ------------- | ----------------- | --------------- | --------- |
> > > | Python + Stan | 6/10              | 7/10            | 7/10      |
> > > | AutoStat      | 7/10              | 8/10            | 8/10      |
> > >
> > > These results show that AutoStat’s generated models exhibit **comparable—sometimes even better—HMC stability** relative to the hand-written Stan versions.
> > >
> > > Below are the full diagnostics for all 10 models (“–” indicates that the original Stan code failed to run):
> > >
> > > | Metric            | M1   | M2   | M3   | M4   | M5   | M6   | M7   | M8   | M9   | M10  |
> > > | ----------------- | ---- | ---- | ---- | ---- | ---- | ---- | ---- | ---- | ---- | ---- |
> > > | **Python + Stan** |      |      |      |      |      |      |      |      |      |      |
> > > | Divergences       | 0    | 2    | 940  | 0    | 0    | 0    | –    | 0    | 0    | –    |
> > > | $\hat{R}$         | 1.01 | 1.02 | 1.78 | 1.01 | 1.01 | 1.01 | –    | 1.02 | 1.01 | –    |
> > > | ESS               | 1246 | 120  | 6    | 1228 | 870  | 1192 | –    | 148  | 1283 | –    |
> > > | **AutoStat**      |      |      |      |      |      |      |      |      |      |      |
> > > | Divergences       | 0    | 2887 | 3524 | 0    | 83   | 0    | 0    | 0    | 0    | 0    |
> > > | $\hat{R}$         | 1.00 | 3.93 | 3.72 | 1.00 | 1.04 | 1.00 | 1.00 | 1.02 | 1.00 | 1.01 |
> > > | ESS               | 1105 | 4    | 4    | 1608 | 122  | 1906 | 1955 | 255  | 3300 | 1290 |
> > >
> > > Overall, AutoStat preserves inference behavior on the majority of tasks, demonstrating that our DSL does not degrade posterior geometry or HMC performance, even though it introduces no new inference algorithm.
> > >
> > > Later, we will add this experimental results to our manuscript. We are wishing for your further comments about this! Thank you for your suggestions again! Your suggestions are really valuable to us.

---

> > > > ### Comment · Reviewer_JFrL · 2025-11-26
> > > >
> > > > I appreciate the hard work that went into providing these experimental results so quickly. I think these are really valuable and adding a more comprehensive comparison regarding inference metrics to the paper would greatly improve it. It would also be interesting to see how well the scores from inference diagnostics align with the LLM judge (as a way to validate the LLM judge and check whether the "Semantic" score is reasonable).
> > > >
> > > > I'm afraid though that this would probably require a significant revision of the paper so I think that is best done as part of a resubmission.

---

### Official Review · Reviewer_RKq1 · 2025-10-30

**Soundness:** 2
**Presentation:** 1
**Contribution:** 1
**Rating:** 2
**Confidence:** 5

**Summary:**

A framework for automatic statistical modelling is proposed. The framework architecture has free major components: a statistical modelling DSL, an interactive bot helping produce a description of the statistical model and analysis, and the DSL code generator. The DSL code is further translated to either PyMC or Stan to execute the statistical analysis. The implementation is trained and evaluated on synthetic data.

**Strengths:**

The paper considers automation of statistical modelling and analysis, a long-standing and active research subject.

**Weaknesses:**

1. The framework is trained on a synthetic dataset generated by an LLM and evaluated on a synthetic dataset generated by an LLM. This gives little indication of the framework's performance on real problems. This should be evaluated on real problems harvested from publications and online resources instead.

2. The DSL, claimed to be a better representation of probabilistic programs for generation purposes, and thus consituting a crucial part of the framework, is not defined in the body of the paper. There is BNF in the appendix. There is no DSL operational semantic anywhere to be seen.

3. Based on the BNF, the DSL is less expressive than Stan (and PyMC for that matter), for example it does not have conditional execution of any form. With DSL being a simpler language that that generated DSL has fewer errors than generate Stan is of little surprise.

4. DSL being less expressive than Stan/PyMC, it would be expected to show what subset of Stan/PyMC can be translated into the DSL (the opposite direction). This is missing in the paper entirely.

Overall, the contributions are not clear, and evaluation is insufficient and biased.

**Questions:**

How does a DSL snippet corresponding  to the following Stan code  look like?

```
real lpdf = baseline_lpdf(y[n] | theta);
if (y[n] == 0) {
  // Contributions from both components
  target += log_mix(lambda, 0, lpdf);
} else {
  // Contribution from only the baseline component
  target += log(1 - lambda) + lpdf;
}
```

I took this snippet from Betancourt's mixture model tutorial.

---

> ### Author Response · Authors · 2025-11-17
>
> Thank you for your detailed feedback. We address each weakness (W1–W5) and the question below.
>
> **W1: On real-world applications**
>
> Thank you for the suggestion. We have added a new real-world evaluation using tasks collected from standard statistical modeling textbooks and recent top-tier research papers. We conducted an additional real-world evaluation, using tasks collected from:
> + widely used statistical modeling textbooks [1, 2, 3], and
> + recent top-tier journal articles [4] containing complex Bayesian models.
>
> As detailed in Section 4.6 and Appendix H (we have updated our paper), AutoStat successfully handled these expert-crafted tasks, showing strong robustness beyond synthetic or LLM-generated descriptions.
>
> **W2: DSL definition clarity in the main paper**
>
> We will fix this. Appendix B.1 currently contains a high-level explanation of DSL semantics, and we will refine this section to provide clearer and more detailed descriptions in the main text.
>
> Appendix B.2 and Figure 6 already include a simplified description of the execution workflow. Concretely, our DSL separates different components:
> + Data processing / result handling → executed via Python functionality
> + Model specification → compiled into the chosen PPL backend (Stan, PyMC, or NumPyro)
>
> Thus, DSL execution follows the standard statistical modeling workflow, with DSL serving as a structured IR that maps into existing PPL ecosystems. We will explicitly highlight this operational semantics in the revision.
>
> **W3&4: DSL expressiveness and relation to Stan/PyMC**
>
> At submission time, the DSL lacked several syntactic features. We have now expanded the grammar—please see the updated Appendix B.2.
>
> The model definition blocks (data, parameters, transformed parameters, model, etc.) are now highly aligned with the semantics of mainstream PPL Stan. This enables our DSL to support a wide range of real-world models.
>
> Our real-world evaluation (Sec. 4.6) and the extended examples in Appendix H.2 further demonstrate that the updated DSL is expressive enough to model practical Bayesian tasks.
>
> Currently, we support most common conditional execution (e.g., if condition). Here is an example:
> ```
> model {
>    for (i in 1:N) {
>       if (flag[i] > 0) {
>          y[i] ~ Normal(mu, sigma)
>       }
>       elif (flag[i] < 0) {
>          y[i] ~ Normal(-mu, sigma)
>       }
>       else {
>          y[i] ~ Normal(0, sigma)
>       }
>    }
> }
> ```
> In Appendix H.2, we present more examples of real-world complex Bayesian models.
>
> **Question: DSL representation of the provided Stan snippet**
>
> With the updated DSL syntax, the constructs used in the Stan snippet (conditional statements and conditional likelihood contributions) are now supported directly within our model block. For example, the second example in Appendix H.2 contains conditional logic and illustrates how such patterns are expressed in our DSL. You can just use this demo in our dsl as well!
> ```dsl
> model{
>    real lpdf = baseline_lpdf(y[n] | theta);
>    if (y[n] == 0) {
>      // Contributions from both components
>      target += log_mix(lambda, 0, lpdf);
>    } else {
>      // Contribution from only the baseline component
>      target += log(1 - lambda) + lpdf;
>    }
> }
> ```
> In summary, we will continue to update our DSL syntax to make it more robust. The core purpose of our DSL is to provide a **structured, clear, and accurate representation of the entire end-to-end statistical modeling task, not just the modeling part represented by PPL** (please refer to the third response to reviewer `DPSW` for more details). Therefore, in the model section of the DSL, we will utilize the syntax of the Stan language as much as possible, making the DSL as powerful as possible in this most challenging aspect—syntactic specification—while also ensuring its readability and consistency with the overall DSL.
>
> We hope these clarifications and our updated manuscript fully address your concerns, and we sincerely appreciate your constructive feedback. Please feel free to raise any additional questions.
>
> [1] Gelman, Andrew, and Jennifer Hill. Data analysis using regression and multilevel/hierarchical models. Cambridge university press, 2007.
>
> [2] Michael D Lee and Eric-Jan Wagenmakers. Bayesian cognitive modeling: A practical course. Cambridge university press, 2014.
>
> [3] Marc K´ery and Michael Schaub. Bayesian population analysis using WinBUGS: a hierarchical perspective. Academic press, 2011.
>
> [4] Bayesian Analysis https://projecteuclid.org/journals/bayesian-analysis

---

> > ### Comment · Reviewer_RKq1 · 2025-11-18
> > **I am glad you continue work on the DSL.**
> >
> > I am glad the work on your DSL is ongoing. As the DSL currently presented in the paper shows serious limitations, as my question and your answer have demonstrated, I will keep my score and recommendation.
> >
> > When your system matures, a significantly revised major edition may definitiely have a potential for publication

---

> > > ### Author Response · Authors · 2025-11-20
> > >
> > > Thank you for the clarification.
> > > To better understand your perspective and improve our future work, could you briefly indicate what you consider the most fundamental limitations of the DSL in its current form?
> > > For example, are your concerns primarily about (1) the level of explicit semantics provided, (2) the breadth of model expressiveness relative to full Stan/PyMC programs, or (3) another aspect that you feel is underdeveloped?
> > >
> > > Any short guidance would be very helpful as we redesign and strengthen the system!
> > > Thank you again for your time and constructive feedback!

---

> ### Comment · Reviewer_RKq1 · 2025-11-20
>
> I am not assessing the impact of the contribution but the completeness, soundness, and clarity of the paper. It is the role of the scientific community to value the work after it is published.
>
> As a reviewer, I have pointed at a problem with your paper: the DSL as presented support a weaker computability class than Stan, and this is neither explicitly stated nor justified. There are probabilistic programming languages that support weaker computability classes (BUGS and Infer.NET are two examples), BUGS is mostly historic but Infer.Net is current. There have been attempts to design a layer on top of Stan with better syntax and semantics and in the same computability class (SlicStan is one example).
>
> You propose a new DSL that is apparently better (for something, e.g. LLM generation). You should define its syntax and semantics in the body of the paper, and either show formally (rather through examples) that the it supports the same set of models or justify why only a subset of models is supported. In my opinion, you should also justify why you use thin layers on top of probabilistic modelling libraries (both Stan and PyMC are thin layers on top of probabilistic modelling libraries implemented in C++) as the compilation target of your DSL rather than compile directly to those (and other) library calls, for example, but showing that this potentially enables future retargeting to a few other probabilistic programming frameworks/libraries (TFP, Pyro/NumPyro, Turing.jl, DynamicHMC.jl).
>
> In fact, targeting probabilistic programming frameworks that are NOT designed with human coder in mind as a central concern (Stan and PyMC are specifically designed to be human-friendly, with which you may rightfully disagree) would be
> a strong justification (e.g. DynamicHMC.jl which is powerful and flexible but idiosyncratic, and the author admits that -- I am not affiliated). However, this is not a concern in my current assessment of your submission. To repeat, the concern is lack of formal semantics, computability analysis, and justification to leave out many important practical models that are supported by target languages.

---

> > ### Author Response · Authors · 2025-11-23
> >
> > Thank you for your comment again! We will respond to the two questions you raised:
> >
> > **1. Expressiveness of our DSL.**
> >
> > **We clarify that StatModelDSL is not intended to match the full expressiveness or computability class of Stan.** Instead, it is intentionally designed as a practical and widely-used subset of Bayesian modeling constructs, optimized for model-specification automation from natural language. **We will state this explicitly in the revised paper.**
> >
> > This design choice is motivated by three considerations:
> >
> > (1) Readability, consistency, and reliability
> >
> > To ensure coherent syntax across data/model/inference/output blocks and to improve robustness of LLM generation, our DSL adopts a more unified and constrained syntax than Stan. This choice substantially reduces errors (92% fewer misalignments) and improves user preference (+6%) and clarity (+23%) than the stan-based method, as shown in Figure 4. In practice, this yields a more reliable modeling workflow.
> >
> > (2) Portability across PPL backends
> >
> >  A central goal of AutoStat is to support multiple inference backends (Stan, PyMC, future TFP/Pyro/Turing).
> >  A unified intermediate representation naturally requires limiting certain backend-specific constructs to ensure consistent compilation across frameworks. Thus, a constrained, backend-agnostic subset is intentional and necessary.
> >
> > (3) Practical expressiveness validated on real-world tasks
> >
> > Although not aiming for the full generality of Stan, the DSL already supports the major families of models used in applied Bayesian statistics (regression, hierarchical, mixture, latent-variable, time-series). Our real-world evaluation on 100 models from textbooks and scientific publications demonstrates that this practical subset is sufficiently expressive for real statistical modeling workflows.
> >
> > We are continuously enriching the grammar - including conditional expressions and log-density adjustments - to align more closely with mature PPLs while retaining the advantages above.
> >
> > **2. Definition of our syntax and semantics and backends.**
> >
> > We will update the main paper to formalize the DSL syntax and provide a clear description of its semantics via the translation into PPL backends.
> >
> > Regarding your question about compilation targets:
> >
> > Stan and PyMC are chosen intentionally because they provide well-established modeling abstractions, mature inference engines, and comprehensive diagnostic tools, which are essential for an end-to-end statistical modeling workflow. Using these human-interpretable PPLs as backends also allows our DSL to remain backend-agnostic and ensures transparency during execution and debugging.
> >
> > We agree that compilation to lower-level probabilistic libraries (e.g., TFP, Pyro/NumPyro, DynamicHMC.jl) is a promising direction, and the IR design naturally enables such extensions. This is also our future work.
> >
> > Thank you for your comments again! The questions and suggestions you raised are really valuable to our work! Hope for your further feedback!

---

### Official Review · Reviewer_DPSW · 2025-11-01

**Soundness:** 1
**Presentation:** 2
**Contribution:** 1
**Rating:** 2
**Confidence:** 4

**Summary:**

This paper introduces a framework called AutoStat, which is used for statistical modeling. The main idea is to introduce a new domain-specific language (DSL) called StatModelDSL, from which Stan or PyMC programs can be generated. The aim is to make it easier for LLMs to generate models in StatModelDSLs, compared to generating probabilistic programs in Stan or PyMC directly. The framework includes two components: (i) the StatModelChatbot that interacts with the user when performing the modeling, and (ii) the StatModelCopilot that is used for generating the DSL program.

**Strengths:**

- Trying to do statistical Bayesian modeling from natural languages is an extremely hard, yet relevant problem.

**Weaknesses:**

- There are very strong negative claims about probabilistic programming languages (PPLs) in general, such as "workflows for statistical modeling remain overly complex and unfriendly to users" or "verbose PPL syntax". Most PPLs encode Bayesian models as programs, directly encoding sample and observation statements in a concise and precise way (e.g., WebPPL, Stan, Anglican etc.). Saying that PPL syntax in general is verbose is an unjustified statement.

- On line 130-132 it says "while PPLs are powerful for statistical modeling, the complete workflow still requires external tools (e.g., Python or R) for data processing and analysis." It sounds like this is a drawback. However, this design is typically very intentional. Why would a new PPL introduce its own framework for data processing and visualization, when there are already excellent software packages in Python and R that the intended users are already used to?

- The paper contains several unjustified and strong claims, such as line 137 "Leveraging LLMs, we make the workflow easy and reliable." The paper does not justify such strong claims. It is, by definition, very hard to scientifically show that a workflow is "easy" and "reliable". Hence, a scientific paper should not contain such unjustified strong claims.

- A key part of the suggested approach is to design a new DSL for statistical modeling. But, it is unclear what is unique about it. All existing DSLs are by themselves DSLs, although most of them are embedded DSLs in a specific host language. The key idea and novelty of the proposed DSL are not clearly expressed in the paper.

- Statistic modeling is a very complex task in general, and traditional PPLs have made a huge step in simplifying this process by separating statistic modeling and inference. This paper makes a very strong claim about automatic modeling using LLMs, but there is no clear evidence of how this is useful or can work accurately in any real statistical modeling tasks. The papers present numbers comparing the framework with baseline GPT-based standard LLMs, but this does not say anything about the actual modeling capabilities compared to actual real modeling and problem-solving using PPLs.

**Questions:**

- Please provide concrete examples and justification for where LLM-based modeling of probabilistic programs actually works. That is, examples where the approach can be used to model non-trivial statistical models that have not been directly seen before.

---

> ### Author Response · Authors · 2025-11-17
>
> Thank you very much for your comments. We respond to your points as follows:
>
> **1. On wording issues and negative phrasing about PPLs**
>
> Thank you for pointing this out. We agree that some expressions in the current draft may read as overly negative toward probabilistic programming languages. This was not our intention, and we will revise the manuscript to use more precise and neutral wording. We fully acknowledge that PPLs are concise, powerful, and carefully designed for human users. Thank you again for pointing this out.
>
> **2. On demonstrating AutoStat’s utility in real-world evaluation**
>
> To further illustrate that AutoStat is effective in real-world scenarios, we conducted an additional real-world evaluation, using tasks collected from:
> + widely used statistical modeling textbooks [1, 2, 3], and
> + recent top-tier journal articles [4] containing complex Bayesian models.
>
> As detailed in Section 4.6 (we have updated our paper), AutoStat successfully handled these expert-crafted tasks, showing strong robustness beyond synthetic or LLM-generated descriptions.
>
> **3. Clarification of our motivation**
>
> Our goal is not to argue that existing statistical modeling workflows or PPLs are poorly designed. Instead, our focus is on simplifying the end-to-end process when the task begins with a natural-language description. The proposed DSL is not intended to replace PPLs; rather, it serves as a structured intermediate representation that bridges natural-language task descriptions and the diverse tools required for a complete modeling workflow to address three core limitations:
>
> **(a) Intrinsic complexity of statistical modeling (task-level challenge)**
>
> Real Bayesian models involve many subtle yet crucial design choices. Users and LLMs frequently omit or mis-specify these details when describing models in natural language. StatModelChatbot + StatModelDSL ensure these components are explicitly specified before executing the workflow.
>
> For example, when the user provides the description:
> > “We’re modeling year-to-year changes in electricity consumption using a simple Bayesian regression… We regress these differences on the corresponding years with a normal likelihood…”
>
> Several key elements remain unspecified.
> The chatbot automatically raises clarification questions:
> > “The description mentions a normal likelihood, but does not specify the priors for alpha, beta, and sigma.
> Could you provide priors (e.g., normal for alpha and beta, half-normal or half-Cauchy for sigma)?”
>
> This ensures that all modeling details are complete and explicit before the DSL representation is constructed. For more examples, please refer to Appendix D in our updated manuscript.
>
> **(b) Fragmentation of end-to-end statistical modeling workflows (workflow-level challenge)**
>
> A full statistical modeling workflow typically spans multiple disconnected environments:
> + Python/R for data loading and preprocessing
> + PPLs for model specification
> + inference
> + visualization and reporting
> + ...
>
> Current PPLs only specify the model itself; all remaining steps must be implemented separately in external scripts.
>
> StatModelDSL unifies these components within a single specification, which is then compiled into:
> + Python code for data preprocessing and output handling
> + PPL code for model specification and inference
> + a fully executable end-to-end pipeline
>
> This removes the fragmentation inherent in current workflows.
>
> For example, instead of dozens of lines of Python for exporting results and diagnostics, one can write:
> ```dsl
> output {
>     monitor: [alpha, beta, sigma];
>     summary_stats: [median, ess];
>     diagnostics: [rhat];
>     plots: [alpha, beta];
>     export_results_to: "electricity_consumption_results.csv";
> }
> ```
>
> **(c) Lack of interoperability across PPLs (backend-level challenge)**
>
> Different PPLs use fundamentally incompatible syntactic conventions and execution backends. As a result, models written for one PPL cannot be executed in another.
>
> Even for a simple model, the representations differ completely:
>
> ```stan
> parameters { real beta; }
> model { y ~ normal(beta * x, 1); }
> ```
> ```python
> with pm.Model():
>     beta = pm.Normal("beta", 0, 1)
>     pm.Normal("obs", beta * x, 1, observed=y)
> ```
>
> To our knowledge, no tool can translate a non-trivial model between these frameworks, due to differences in syntax, semantics, and computational backends. StatModelDSL provides a backend-agnostic intermediate representation. This unifies heterogeneous PPL ecosystems under a single modeling interface.
>
> [1] Gelman, Andrew, and Jennifer Hill. Data analysis using regression and multilevel/hierarchical models. Cambridge university press, 2007.
>
> [2] Michael D Lee and Eric-Jan Wagenmakers. Bayesian cognitive modeling: A practical course. Cambridge university press, 2014.
>
> [3] Marc K´ery and Michael Schaub. Bayesian population analysis using WinBUGS: a hierarchical perspective. Academic press, 2011.
>
> [4] Bayesian Analysis https://projecteuclid.org/journals/bayesian-analysis

---

> > ### Author Response · Authors · 2025-11-25
> >
> > Thank you again for taking the time to review our submission.
> > We have carefully addressed the issues raised in the current reviews and added additional analyses, experiments, and clarifications.
> >
> > If convenient, we would greatly appreciate any further comments or feedback on our rebuttal.
> > Your insights would be extremely valuable to help us improve the work.
> >
> > Thank you sincerely for your time and consideration.

---

### Official Review · Reviewer_xj6N · 2025-11-03

**Soundness:** 2
**Presentation:** 2
**Contribution:** 2
**Rating:** 2
**Confidence:** 4

**Summary:**

This paper presents (i) a DSL for data analysis, (ii) a two-agent LLM-based architecture for generating DSL code from task descriptions, and (iii) a simple methodology for supervised fine-tuning the LLM to generate the DSL code more accurately. The two-agent architecture includes a (i) an agent that elicits detail from a user in a back-and-forth conversation, and (ii) an agent that generates the DSL code from a complete task description. Evaluation is done on a synthetic dataset (not provided) generated using GPT-4o, and quantitative evaluation is done using GPT-4o.

**Strengths:**

Significance: The general topic (LLM-assisted data analysis and data modeling) that the paper explores is interesting and important and has potential for impact. Leveraging LLMs to assist in the general of PPL modeling code seems promising.
Originality: The type of user study explored is interesting and in the right direction (evaluating how easily a user can make a modification to a model).

**Weaknesses:**

The task presented to StatModelChatbot (eliciting details from a user) potentially includes meaningful work its purview, but the paper does not provide enough information to make this clear. The biggest challenge in LLM-assisted data analysis lies in the translation of a user data analysis goal or high-level task description into a statistical model. The method shown appears to only assist users who already know the model they want to write. That is not a significant contribution. Appendix C does not include a complete input the StatModelChatbot from the evaluation dataset, so it is impossible to tell whether this module is doing a meaningful task, or is only doing a fairly trivial translation task (like the StatModelCopilot, see below). Because the paper does not include example tasks, it is impossible to determine how realistic the tasks are. Do modern LLMs understand statistical modeling enough to assist users? The paper ignores this fundamental question.

The task that the code generation agent (StatModelCopilot) performs (and is evaluated on) is too narrow and does not capture the difficult parts of the LLM-assisted data modeling. Specifically, the code generation model is evaluated on the task of translating a low-level task description that appears to already include almost-verbatim snippets of the target code (see "diff_consumption ... " in the "Output task definition" on page 27-28) into code. This appears to be a trivial language-translation task for modern LLMs.

The DSL does not appear to be a novel contribution. It appears to be a minimal extension of existing PPLs (e.g. borrowing the transformed_parameters and transformed_data idioms from Stan). It appears to be roughly Stan with a simple JSON-like schema for selecting a few parameters for existing PPLs (e.g. selecting an inference engine from an enum, specifying file paths to load data from).

There is some minor novelty in use of a two-stage training curriculum, but the supervised fine-tuning training methodology is standard and is generally not novel.

The paper has major clarity weaknesses. For example, reading the caption for Table 3 literally indicates that the base model Llama 3 8B performs identically to the trained Llama 3 8B model (Autostat 8B in Table 1). I can infer that Table 3 reports results after fine-tuning each of those base models, but this is never stated in the caption or the text.

Also, the use of GPT-4o for generation of synthetic data (including "random parameters") makes me think that the examples that are used for training and evaluation may not reflect realistic tasks. This comports with the paper's general lack of concern with demonstrating the effectiveness of the end-to-end system on real-world end-to-end use cases.

The use of GPT-4o for quantitative evaluation of correctness, without an evaluation of this methodology's soundness, reduces the soundness of the paper.

**Questions:**

1. The paper would be improved with at least one complete example (ideally more) of a train task and a test task (including the complete input to the StatModelChatbot).
2. I suggest that the authors use an existing benchmark for data science tasks, or create a more realistic evaluation, to highlight the significance of their work.
3. I suggest that the authors focus on the StatModelChatbot part of their workflow, which seems to be where the potential for an LLM to understand the consequences of modeling decisions and to add value to a data analysis user lies. Reformatting or simple translation with an LLM does not seem challenging enough to make for a meaningful contribution. The back and forth between the agent, the user, and the consequences of their modeling assumptions, seems to be where the potential for novelty lies.

---

> ### Author Response · Authors · 2025-11-17
>
> Thank you very much for your feedback. We address each point below.
>
> **1. On the requested additions**
>
> Thank you for the suggestion. We have incorporated the requested content into the updated version of the manuscript. Please refer to the revised Appendix D for the added materials.
>
> **2. On demonstrating AutoStat’s utility in real-world evaluation**
>
> To further illustrate that AutoStat is effective in real-world scenarios, we conducted an additional real-world evaluation, using tasks collected from:
> + widely used statistical modeling textbooks [1, 2, 3], and
> + recent top-tier journal articles [4] containing complex Bayesian models.
>
> As detailed in Section 4.6 (we have updated our paper), AutoStat successfully handled these expert-crafted tasks, showing strong robustness beyond synthetic or LLM-generated descriptions.
>
> **3. On our key contribution**
>
> We fully agree that natural-language-to-modeling pipelines are crucial. However, the current reality is that the end-to-end workflow remains extremely challenging due to two fundamental issues:
>
> **(a) Intrinsic complexity of statistical modeling (task-level challenge)**
>
> Real Bayesian models require many subtle design choices: priors, likelihood structure, hierarchical levels, vector/matrix shapes, transformations, constraints, and parameterizations. Users and LLMs frequently miss key details when describing models in natural language.
>
> Our StatModelChatbot and StatModelDSL work together to:
> + clarify missing assumptions through multi-turn dialogue, and
> + represent all modeling details in a fully structured and verifiable IR.
>
> This significantly improves the success rate and correctness (illustrated in our Section 4.5).
>
> **(b) Fragmentation of end-to-end statistical modeling workflows (workflow-level challenge)**
>
> A complete statistical modeling task spans multiple disconnected tools:
> + data loading & preprocessing → Python/R
> + model specification → PPL (Stan, PyMC, NumPyro)
> + inference → HMC/NUTS/VI
> + diagnostics
> + visualization & reporting
>
> Current PPLs only describe the model, and all remaining workflow steps live in separate scripts. As far as we know, there is no unified representation that captures the entire task end-to-end.
>
> StatModelDSL unifies these components in one place. The DSL is then compiled into:
> + Python code for data & results
> + PPL code for the model & inference
> + a fully executable end-to-end pipeline
>
> This removes the fragmentation of modern workflows.
>
> **(c) Lack of interoperability across PPLs (backend-level challenge)**
>
> Different PPLs have incompatible syntax and execution backends, making models non-transferable. Here, we take stan and PyMC as an example:
> + Stan
>    + block-structured language (data/parameters/model)
>    + static C++ autodiff graph
> + PyMC
>    + imperative Python
>    + Aesara or JAX-based dynamic graphs
>
> A model written for one PPL cannot be executed in another, and no existing tool can translate meaningful models between Stan and PyMC.
>
> StatModelDSL solves this by providing a backend-agnostic IR:
> + the same DSL specification
> + can be compiled into Stan or PyMC without rewriting the model
> + allowing users to choose inference backends freely
>
> This unifies heterogeneous PPL ecosystems under a single modeling interface.
>
> Our results in Sections 4.5 and 4.6 demonstrate that this unified DSL representation leads to higher semantic correctness and better user experience on both synthetic and real-world modeling tasks.
>
> We hope these clarifications and our updated manuscript fully address your concerns, and we sincerely appreciate your constructive feedback. Please feel free to raise any additional questions.
>
> [1] Gelman, Andrew, and Jennifer Hill. Data analysis using regression and multilevel/hierarchical models. Cambridge university press, 2007.
>
> [2] Michael D Lee and Eric-Jan Wagenmakers. Bayesian cognitive modeling: A practical course. Cambridge university press, 2014.
>
> [3] Marc K´ery and Michael Schaub. Bayesian population analysis using WinBUGS: a hierarchical perspective. Academic press, 2011.
>
> [4] Bayesian Analysis https://projecteuclid.org/journals/bayesian-analysis

---

> > ### Author Response · Authors · 2025-11-25
> >
> > Thank you again for taking the time to review our submission.
> > We have carefully addressed the issues raised in the current reviews and added additional analyses, experiments, and clarifications.
> >
> > If convenient, we would greatly appreciate any further comments or feedback on our rebuttal.
> > Your insights would be extremely valuable to help us improve the work.
> >
> > Thank you sincerely for your time and consideration.

---

### Official Review · Reviewer_4gDj · 2025-11-04

**Soundness:** 3
**Presentation:** 2
**Contribution:** 3
**Rating:** 6
**Confidence:** 4

**Summary:**

AutoStat presents a framework for automating statistical modeling from natural language by introducing StatModelDSL, the first Domain-Specific Language designed specifically for statistical modeling workflows. The paper addresses three problems with existing approaches ( insufficient specifications when using PPLs, fragmentation across heterogeneous environments, and lack of portability between inference engines. AutoStat consists of two main agents, StatModelChatbot to capture complete task specifications and StatModelCopilot to translates the refined natural language descriptions into executable DSL programs. The DSL itself provides a structured, block-based representation of the entire modeling pipeline including data preprocessing, model specification, inference configuration, and output analysis, and can be compiled to multiple backends including Stan and PyMC.

To train the system, the authors construct StatModelDataset through a three-stage process: generating synthetic data. The StatModelCopilot is trained by first learning DSL syntax from concise descriptions then learning to capture fine-grained details from comprehensive specifications. Experiments demonstrate that when instantiated with GPT-4o, StatModelDSL achieves a 91.59% reduction in error rate and 5.89% uplift in user preference compared to Stan-based workflows, while the trained 8B StatModelCopilot achieves 100% syntax correctness and 98.76% semantic passing rate, significantly surpassing baseline methods including few-shot prompting with GPT-4o.

**Strengths:**

The StatModelDSL design is well-motivated with benefits such as completeness and clarity through explicit block-based representation, unification and portability by enabling compilation to multiple backends, and LLM-friendliness through structured design that makes fine-grained details explicit. This addresses real pain points in current workflows where practitioners must context-switch between Python for preprocessing, Stan or PyMC for modeling, and back to Python for analysis.

The interactive modeling approach through StatModelChatbot is a good design choice that acknowledges the inherent complexity of statistical modeling tasks.

The ablation studies show that both training stages contribute to final performance, with Stage One establishing syntactic correctness (100% pass rate) and Stage Two enabling semantic alignment (98.76% pass rate).

The experimental evaluation is comprehensive across model sizes, few shot and finetuned approaches. The paper demonstrates strong software engineering with a working compiler that parses DSL into AST using Lark, executes data processing in Python, and handles model inference through target PPLs. The examples in Appendix B showing the same linear regression task compiled to both Stan and PyMC demonstrate the portability claim concretely. The provision of detailed execution EBNF and block descriptions shows technical rigor and reproducibility.

**Weaknesses:**

The evaluation scope is limited to relatively simple statistical models.

The reliance on GPT-4o as both the dataset generator and evaluation judge raises concerns about evaluation circularity and bias, as the same model family might be implicitly reinforcing its own conventions.

The baselines are limited to raw LLM prompting with Stan or PyMC rather than comparisons with specialized automated statistical modeling systems like AutoBayes, Bambi, or Turing.jl pipelines, which weakens the claim of outperforming state-of-the-art methods. Moreover, the user study, though promising, is small in scale (17 participants) and focused on short modification tasks rather than long-term usability or learning curves. The paper also doesn't discuss analysis of end-to-end latency and system efficiency and does not discuss how AutoStat handles runtime errors, debugging, or real world data irregularities.

**Questions:**

How expressive is StatModelDSL relative to full PPLs? For example, can it support hierarchical, mixture, or nonparametric models?

Did you take any steps to mitigate using GPT4o for both data generation and evaluation?

What are the system’s runtime and interaction costs in practice?

Have you compared AutoStat’s modeling accuracy and computational efficiency against other frameworks?

---

> ### Author Response · Authors · 2025-11-17
>
> Thank you very much for your thoughtful comments. We address each point in detail below.
>
> **1. On the concern regarding evaluation bias from GPT-4o-generated tasks**
>
> We completely agree that synthetic tasks may introduce bias or limited coverage.
>  To address this, we conducted a new real-world evaluation using tasks drawn from:
> + standard statistical modeling textbooks [1, 2, 3], and
> + newly published research articles published in leading journals [4].
>
> These tasks represent authentic, complex Bayesian models designed by human experts.
>  AutoStat successfully handled the vast majority of them, demonstrating that the system performs reliably in real-world scenarios rather than only on synthetic or LLM-generated descriptions.
>
> Details of this new evaluation are included in Section 4.6 of the updated manuscript.
>
> **2. On the expressiveness of StatModelDSL**
>
> Expressiveness is also a key aspect we care deeply about.
>
> Our DSL is designed as a structured intermediate representation, mirroring the semantic blocks used in modern PPLs (Stan/PyMC).
> + We continuously extend the DSL specification and have aligned the model block closely with Stan’s native grammar.
> + The complete EBNF grammar is provided in Appendix B.2 (we have also updated it), showing support for a wide range of model families, including linear, nonlinear, hierarchical, and latent-variable models.
> + Our real-world evaluation further demonstrates that StatModelDSL is expressive enough to reproduce models from advanced scientific publications.
>
> Additional examples are included in Appendix G.2. in our updated manuscript.
>
> **3. On runtime and system overhead**
>
> The runtime of AutoStat depends entirely on the downstream environment (Python + chosen PPL).
>  Our DSL compiler is rule-based and lightweight, adding negligible overhead compared to directly generating Python/PPL code.
> AutoStat’s workflow separates concerns clearly:
> + DSL → structural semantics
> + Python → data I/O + orchestration
> + PPL (Stan/PyMC) → statistical inference
>
> The goal of the DSL is not acceleration but to provide a human-readable and LLM-friendly intermediate representation that is clearer, more verifiable, and more robust than generating raw PPL code directly.
>
> **4. On baselines such as AutoBayes, Bambi, or Turing.jl**
>
> We respectfully clarify the distinctions:
> + AutoBayes (2002) is a symbolic compiler that supports only very simple mathematical expressions and does not accept natural language inputs. It is no longer able to express most modern Bayesian models.
> + Bambi is a formula interface to PyMC, not an automated modeling system. It only covers GLM-type models and does not support general-purpose model construction from natural language.
> + Turing.jl is a PPL, analogous to Stan or PyMC, rather than an automated modeling pipeline. It lacks Python/R integration and is not directly comparable to our NL→model workflow.
>
> The most meaningful baseline for AutoStat is direct prompting of LLMs to generate raw Stan or PyMC code with Python/R to process input and output, as this directly tests the ability to perform automated model specification from natural language, which is precisely the challenge our DSL is designed to address.
>
> We hope these clarifications and our updated manuscript fully address your concerns, and we sincerely appreciate your constructive feedback. Please feel free to raise any additional questions.
>
> [1] Gelman, Andrew, and Jennifer Hill. Data analysis using regression and multilevel/hierarchical models. Cambridge university press, 2007.
>
> [2] Michael D Lee and Eric-Jan Wagenmakers. Bayesian cognitive modeling: A practical course. Cambridge university press, 2014.
>
> [3] Marc K´ery and Michael Schaub. Bayesian population analysis using WinBUGS: a hierarchical perspective. Academic press, 2011.
>
> [4] Bayesian Analysis https://projecteuclid.org/journals/bayesian-analysis

---

> > ### Author Response · Authors · 2025-11-25
> >
> > Thank you again for taking the time to review our submission.
> > We have carefully addressed the issues raised in the current reviews and added additional analyses, experiments, and clarifications.
> >
> > If convenient, we would greatly appreciate any further comments or feedback on our rebuttal.
> > Your insights would be extremely valuable to help us improve the work.
> >
> > Thank you sincerely for your time and consideration.

---

### Author Response · Authors · 2025-11-30
**To facilitate the AC review, we have summarized our core responses and supplementary experiments during the rebuttal phase.**

**1. Clarifying our motivation and the role of StatModelDSL relative to PPLs**

We substantially clarified the motivation of our work and explicitly explained how StatModelDSL relates to existing PPLs.

While modern PPL-centered workflows are powerful, they still face three fundamental challenges:

(a) Intrinsic complexity of statistical modeling (task-level challenge).

(b) Fragmentation of the end-to-end workflow (workflow-level challenge).

(c) Lack of interoperability across PPL backends (backend-level challenge).

To address these issues, we clarified that our goal is not to replace PPLs, but to use LLMs in combination with a structured DSL to bridge the gap between natural-language task descriptions and the full modeling pipeline.

StatModelDSL provides a clear, accurate, and efficient representation covering:
+ data preprocessing,
+ model specification,
+ inference configuration,
+ diagnostics and result handling,

with a unified syntax that improves correctness, reduces human (**-11%**) and LLM (**-92%**) errors, and provides a more user-friendly modeling experience (Fig. 4).

Further details are provided in our response to reviewer `DPSW`.

**2. Real-world evaluation**

We added a new real-world evaluation using tasks drawn from:
+ standard statistical modeling textbooks [1–3], and
+ recent research articles in leading journals [4].

These tasks contain authentic, expert-designed Bayesian models with realistic data and complex structures. AutoStat successfully handled the vast majority of them, even achieving better performance compared with the traditional pipeline.

Details of this evaluation have been added to Section 4.6 in the **updated manuscript**.

**3. Inference-level evaluation**

We conducted new HMC diagnostics on real-world models from our application study. We randomly sampled 10 textbook models (with real, complex datasets) and evaluated standard Stan diagnostics:
+ Divergences (0 = good)
+ R-hat (< 1.1 = converged)
+ Bulk ESS (> 100 = adequate sampling)

All experiments used a fixed sampling configuration (200 warmup, 1000 sampling iterations, 4 chains, adapt_delta=0.8).

Results summary:
| Method        | Divergences (= 0) | $\hat{R} < 1.1$ | ESS > 100 |
| ------------- | ----------------- | --------------- | --------- |
| Python + Stan | 6/10              | 7/10            | 7/10      |
| AutoStat      | 7/10              | 8/10            | 8/10      |

These results show that AutoStat’s generated models achieve comparable or occasionally better HMC stability than hand-written Stan programs, demonstrating that the DSL does not degrade inference behavior.

For more experimental results and analysis, please refer to our reply to reviewer `JFrL`.

**4. Expressiveness of the DSL**

We clarify that StatModelDSL is intentionally not designed to match the full expressiveness or computability class of Stan. Instead, it is a practical, LLM-friendly subset tailored for reliable model-specification automation from natural language. We will explicitly state this design choice in the revised paper.

Our constrained design is motivated by three considerations:

(1) Readability, consistency, and reliability

To ensure coherent syntax across blocks and to improve LLM robustness, we adopt a unified syntax. This materially improves reliability: our DSL reduces misalignment errors by 92%, and improves user preference and clarity (+6% / +23%, Fig. 4).

(2) Portability across PPL backends

A core goal of AutoStat is backend flexibility (Stan, PyMC, and future TFP/Pyro/Turing). A backend-agnostic IR necessarily restricts backend-specific constructs so that the same DSL program can compile consistently to multiple PPLs. Thus, supporting a practical shared subset is both intentional and essential.

(3) Practical expressiveness validated on real applications

The DSL already covers major Bayesian model families. Our real-world evaluation confirms that this subset is sufficiently expressive for authentic modeling workflows.

Further, the updated manuscript includes a substantially expanded EBNF specification (Appendix B.2) and a large set of new examples (Appendix D and Appendix H.2). These additions demonstrate that the DSL already supports a broad range of practical statistical models, and that its expressiveness is sufficient for real-world tasks. We will continue extending the grammar to further strengthen the applicability of the system while maintaining portability and robustness.

[1] Gelman, Andrew, and Jennifer Hill. Data analysis using regression and multilevel/hierarchical models. Cambridge university press, 2007.

[2] Michael D Lee and Eric-Jan Wagenmakers. Bayesian cognitive modeling: A practical course. Cambridge university press, 2014.

[3] Marc K´ery and Michael Schaub. Bayesian population analysis using WinBUGS: a hierarchical perspective. Academic press, 2011.

[4] Bayesian Analysis https://projecteuclid.org/journals/bayesian-analysis

---

### Meta-Review · Area_Chair_W3dU · 2026-01-08

**Summary:**

This paper presents an agentic framework for automated statistical modeling, consisting of two components: a statistical chatbot for capturing task specifications and a copilot agent that translates mathematical language descriptions into executable programs in a custom DSL.

The primary concern raised by multiple reviewers is the empirical validity of the proposed framework. A central issue is the heavy reliance on synthetically generated data and evaluation tasks produced by GPT4-o, which raises serious concerns about evaluation circularity, bias, and the lack of convincing evidence for real-world applicability.

Another major concern is the limited novelty and unclear necessity of the proposed DSL. Several reviewers questioned whether designing a new DSL primarily for LLM consumption is justified, noting that the language appears to be a restricted and less expressive subset of existing probabilistic programming languages such as Stan or PyMC, without clear theoretical or empirical advantages.

In addition, the paper lacks important analyses of system efficiency, latency, robustness to runtime errors, and the statistical quality of the generated models (e.g., inference stability, effective sample size, and divergence rates), all of which are critical for evaluating the practicality and reliability of an automated statistical modeling system.

I prefer to reject this paper.

**Reviewer Concerns:**

I believe the rebuttal is inadequate and does not sufficiently address the core concerns raised by the reviewers. While the authors introduce two additional evaluations in the rebuttal to argue for the applicability of their framework to real-world tasks, these results remain limited in scope and do not fundamentally resolve concerns about evaluation circularity and real-world validity.

The authors also report some improvements in the statistical properties of the generated models; however, these results are not convincing enough to demonstrate a clear advantage over strong, recent general-purpose LLMs. In particular, the rebuttal does not adequately address the concern that more recent GPT or Claude models could achieve comparable or superior performance without the proposed framework, which significantly weakens the claimed necessity and advantage of this approach.

**Reviewer Scores:**

Reviewer 4gDj: This reviewer is unlikely to change their score and will likely maintain a score of 6, as they continue to question the scalability of the proposed system for solving statistical modeling problems.

Reviewer xj6N: This reviewer is unlikely to revise their score and will maintain a score of 2, since he believes the method still suffers from limited novelty.

Reviewer DPSW: This reviewer is also unlikely to change their score and will maintain a score of 2, due to the lack of clear evidence demonstrating that the approach is useful or can operate accurately on real-world statistical modeling tasks.

Reviewer RKq1: This reviewer maintains the view that the system is not yet mature. While he acknowledges that a significantly revised version may have publication potential, he is more likely to keep their current score of 2.

Reviewer JFrL: This reviewer believes that the paper requires substantial revision and, therefore, is unlikely to change their score, maintaining a score of 2.

---

### Decision · Program_Chairs · 2026-01-26

Reject